# Tri-Level Navigator: LLM-Empowered Tri-Level Learning for Time Series OOD Generalization

**Chengtao Jian**
Tongji University, Shanghai, China
`jct@tongji.edu.cn`

**Kai Yang**[*]
Tongji University, Shanghai, China
`kaiyang@tongji.edu.cn`

**Yang Jiao**
Tongji University, Shanghai, China
`yangjiao@tongji.edu.cn`

## Abstract

Out-of-Distribution (OOD) generalization in machine learning is a burgeoning area of study. Its primary goal is to enhance the adaptability and resilience of machine learning models when faced with new, unseen, and potentially adversarial data that significantly diverges from their original training datasets. In this paper, we investigate time series OOD generalization via pre-trained Large Language Models (LLMs). We first propose a novel **T**ri-level learning framework for **T**ime **S**eries **O**OD generalization, termed TTSO, which considers both sample-level and group-level uncertainties. This formula offers a fresh theoretic perspective for formulating and analyzing OOD generalization problem. In addition, we provide a theoretical analysis to justify this method is well motivated. We then develop a stratified localization algorithm tailored for this tri-level optimization problem, theoretically demonstrating the guaranteed convergence of the proposed algorithm. Our analysis also reveals that the iteration complexity to obtain an $\epsilon$-stationary point is bounded by $\mathcal{O}(\frac{1}{\epsilon^2})$. Extensive experiments on real-world datasets have been conducted to elucidate the effectiveness of the proposed method.

## 1 Introduction

In machine learning, a common challenge arises when the distributions of training and test sets differ significantly [Quiñonero-Candela et al., 2009]. This mismatch demands that models, trained on specific distribution data, should generalize well on unseen distribution data, known as OOD generalization [Shen et al., 2021, Zhou et al., 2022]. Despite a vast amount of research on the OOD generalization [Zhang et al., 2017, Sagawa et al., 2019, Huang et al., 2020, Arjovsky et al., 2019], the field of OOD generalization in time series is relatively limited and presents more significant challenges. This is primarily due to the inherent temporal dependencies and dynamic changes characteristic of time series data [Hamilton, 2020]. Therefore, a critical aspect of improving time series OOD generalization is to learn robust representations that remain stable despite shifts in distributions.

Recently, the field of machine learning has witnessed remarkable advancements in pre-trained foundation models, with notable examples including Large Language Models (LLMs) such as GPT [Radford et al., 2018], LLaMA [Touvron et al., 2023] and CLIP [Radford et al., 2021]. These models have been instrumental in capturing and leveraging complex patterns across various domains. In addition, using foundation models, especially LLMs, in processing non-linguistic data, e.g., time series is increasingly drawing attention. By fine-tuning only a few handful of parameters, these

---

[*]Corresponding author.

38th Conference on Neural Information Processing Systems (NeurIPS 2024).

models show remarkable versatility in diverse data formats ranging from audio [Ghosal et al., 2023], image [Lu et al., 2021] and time series [Chang et al., 2023, Jin et al., 2023]. Studies indicate that LLMs, as part of the broader foundation model spectrum, demonstrate sophisticated reasoning and strong pattern recognition capabilities [Wang et al., 2023, Chu et al., 2023], fundamentally acting as pattern machines [Mirchandani et al., 2023]. Moreover, LLMs have been shown to be effective in transfer learning across various modalities, due to their data-independent self-attention mechanism [Zhou et al., 2023].

Additionally, recent advancements in vision-language foundation models have shown promising developments in OOD generalization [Zheng et al., 2022], yet the exploration in time series remains underdeveloped. The potential of using foundational models is highlighted by the study in Liu et al. [2023a], Hendrycks et al. [2020], which suggests that pre-trained transformers can improve OOD robustness. *Despite existing efforts, the limited exploration of foundational model applications in time series OOD generalization suggests an emerging field.*

In this paper, we propose a **T**ri-level learning framework for **T**ime **S**eries **O**OD generalization, named TTSO. Unlike conventional OOD generalization methods that focus solely on group-level [Jiao et al., 2022a, Huang et al., 2020] or sample-level uncertainties [Zhang et al., 2017, Zhou et al., 2021, Han et al., 2024], our framework uniquely addresses both by combing a minimization problem for optimal model parameter learning, a maximization problem for dynamically data re-grouping, and another maximization problem for data augmentation under a tri-level framework. To tackle this tri-level problem, we propose a stratified localization algorithm via cutting planes. Leveraging the advanced representation learning capabilities of LLMs, we adapt this tri-level learning framework for fine-tuning LLMs.

Our contributions can be summarized as follows:

- **Tri-level Learning Framework.** In contrast to most existing works in OOD generalization, which primarily focus on either group-level or sample-level uncertainties, TTSO uniquely integrates both aspects under a tri-level learning framework. Specially, this comprehensive framework emphasizes the interdependent relationship between problems of each level, advancing beyond the typical single or bi-level methodologies in OOD generalization. Moreover, a theoretical framework based on Vapnik-Chervonenkis dimension has been developed to rigorously analyze and elucidate the generalization properties of TTSO. We then leverage this tri-level framework to fine-tune LLMs, achieving an maximum 4.9% improvement in performance on time series classification in OOD scenarios.

- **Stratified Localization Algorithm.** To tackle the aforementioned tri-level optimization problem, we develop a stratified localization method using cutting planes. Unlike traditional gradient-based methods, TTSO removes the necessity of computing the hypergradient for the outer optimization problem. This computation is typically very challenging and computationally intensive due to the nested structure of the tri-level optimization problem. Furthermore, the decomposable nature of cutting planes offers a promising avenue for enabling distributed implementations of TTSO, thereby potentially enhancing scalability and computational efficiency.

- **Iteration Complexity Analysis.** To validate the effectiveness of our method, we conducted a thorough theoretical analysis of the algorithm. We theoretically derive that the iteration complexity of the proposed algorithm for achieving an $\epsilon$-stationary point is bounded by $\mathcal{O}\left(\frac{1}{\epsilon^2}\right)$.

## 2  Related Work

In this section, we provide an overview of the foundational concepts and methodologies related to our research, including OOD Generalization and the LLM in time series.

**OOD Generalization**. OOD Generalization research focuses on improving the model's ability to generalize when there is a difference in distribution between the training and test data, and has been widely studied in the fields of Computer Vision (CV) [Recht et al., 2019, Salman et al., 2021] and Natural Language Processing (NLP) [Tu et al., 2020, Schneider et al., 2020]. Existing works for out-of-distribution (OOD) generalization are diverse and can generally be categorized into approaches that consider sample-level [Zhang et al., 2017, Zhou et al., 2021] or group-level [Sagawa et al., 2019,

Huang et al., 2020] uncertainty. However, the exploration of OOD generalization specially for time series remains relatively underdeveloped. A recent study [Lu et al., 2023] introduced 'Diversify', an innovative approach that models time series data from the perspective of distribution and obtain superior performance. In our work, we consider both sample-level and group-level uncertainties and formulate them as a tri-level optimization problem.

**LLM in Time Series**. The integration of LLMs in time series analysis is a rapidly evolving field, drawing significant interest due to their superior pattern recognition and reasoning abilities [Wang et al., 2023, Chu et al., 2023]. A recent example is Time-LLM [Jin et al., 2023], which introduces an innovative method by reprogramming time series and incorporating linguistic prompts, effectively activating the extensive capabilities of LLM. In addition, the OFA framework [Zhou et al., 2023], utilizing the frozen pretrained transformer framework, validates the ersatility and effectiveness of pre-trained models in time series analysis. Another innovative approach is PromptCast [Xue and Salim, 2023], which employs a prompt-based learning method, transforming numerical input and output data into prompts for effective forecasting in zero-shot settings. The TEMPO [Cao et al., 2023] adapts to changes in time series distribution by decomposing time series and adding different prompts for each component and obtain competitive performance in time series forecasting. In specialized domains like traffic [Xue et al., 2022], finance [Zhang et al., 2023] and healthcare [Liu et al., 2023b], LLMs have also shown unique advantages. In this work, we aim to enhance OOD robusness for time series by fine-tuning LLMs with TTSO.

## 3    Problem Formulation and Algorithm

**Notations**. $\mathcal{X}$ and $\mathcal{Y}$ represent the input and target spaces of samples, respectively. The predictor $f_\varphi = h_\omega \circ r_\theta$ consists of the representation function $r_\theta(\cdot)$ with parameter $\boldsymbol{\theta}$ and the classifier $h_\omega$ with parameter $\boldsymbol{\omega}$. The function $f_\varphi : \mathcal{X} \to \mathcal{Y}$ maps time series $\boldsymbol{X} \in \mathcal{X}$ to $Y \in \mathcal{Y}$, where $\boldsymbol{X} \in \mathbb{R}^{T \times F}$ and $Y \in \mathbb{R}_+$, with $T$ as the time series length and $F$ as the feature dimensions. The multivariate time series $\boldsymbol{X}$, composed of $F$ univariate time series each with $T$ observations, is sampled i.i.d. from distribution $\mathbb{P}$ and represented as $\boldsymbol{X} = [\boldsymbol{x}_1, \boldsymbol{x}_2, \ldots, \boldsymbol{x}_F]$, where $\boldsymbol{x}_i = [x_{v_1}, x_{v_2}, \ldots, x_T]$ for $i = 1, \ldots, F$. Assume source domain distributions are $\mathbb{P}_{S_i}$ for $i \in \{1, 2, \ldots, K\}$ and the target domain distribution is $\mathbb{P}_T$. The source domain data $\mathcal{D}_{S_i}$ is sampled i.i.d. from $\mathbb{P}_{S_i}$, and the target domain data $\mathcal{D}_T$ is sampled i.i.d. from $\mathbb{P}_T$.

### 3.1    Preliminary

Given a training dataset $\mathcal{D}_{\text{train}} = \{(\boldsymbol{X_i}, Y_i)\}_{i=1}^{N}$ sampled from the distribution $\mathbb{P}_{\text{train}}(\boldsymbol{X}, Y)$. In supervised learning, the goal is to learn an optimal predictor $f_{\varphi^*}$ on $\mathcal{D}_{\text{train}}$ such that $f_{\varphi^*}$ generalizes well on a test dataset $\mathcal{D}_{\text{test}}$ sampled from the distribution $\mathbb{P}_{\text{test}}(\boldsymbol{X}, Y)$. In self-supervised contrastive learning, for a given time series $\boldsymbol{X}$, we generate two augmented views, $\boldsymbol{X}_{a_1}$ and $\boldsymbol{X}_{a_2}$, using augmentation methods $a_1, a_2 \in \mathcal{A}$. These augmentations produce the time series representations $\boldsymbol{R}_1 = r_\theta(a_1(\boldsymbol{X})) \in \mathbb{R}^{T \times M}$ and $\boldsymbol{R}_2 = r_\theta(a_2(\boldsymbol{X})) \in \mathbb{R}^{T \times M}$, through the representation function $r_\theta$. The objective of contrastive learning is to minimize the distance between positive pairs $(\boldsymbol{R}_1, \boldsymbol{R}_2)$ while maximizing the distance between positive and negative pairs. The general formula for contrastive loss, as detailed in Zhao et al. [2022], is formulated as follows

$$\ell_{\text{con}} = \ell_{\text{align}}(r_\theta; \mathbb{P}, \pi) + \lambda \ell_{\text{reg}}(r_\theta; \mathbb{P}, \pi). \tag{1}$$

The first term aims to minimize the distance between positive pairs in the latent space, while the second term served as a regularizer prevents representation collapse. To evaluate the performance of the model, a classifier $h_\omega$ is trained using the representation function $r_{\theta^*}$

$$h_{\omega^*} = \arg\min_{h_\omega} \mathbb{E}_{(\boldsymbol{X}, Y) \sim \mathbb{P}_{\text{train}}} \ell_{\text{sup}}(h_\omega \circ r_{\theta^*}(\boldsymbol{X}), Y), \tag{2}$$

where the representation function $r_{\theta^*}(\cdot)$ is optimized via the supervised loss in Eq. (2). The classification is performed using $f_{\varphi^*} = h_{\omega^*} \circ r_{\theta^*}$. However, the discrepancy between the training distribution $\mathbb{P}_{\text{train}}(\boldsymbol{X}, Y)$ and the test distribution $\mathbb{P}_{\text{test}}(\boldsymbol{X}, Y)$ poses a challenge for generalizing $f_{\varphi^*}$ to test data. Directly optimizing $\ell_{\text{sup}}(f_\varphi(\boldsymbol{X}), Y)$ may lead to overfitting, compromising performance on unseen data. To mitigate this issue, invariant representation learning [Arjovsky et al., 2019] is employed to handle distribution shifts by learning robust invariant representations across diverse distributions. To achieve this, we begin with the following assumption.

**Assumption 1** (Invariant Assumption [Zhao et al., 2022]). *Considering K different environments (domains) $\mathcal{E}$, there exists a random variable $\psi(\boldsymbol{X})$ such that for any $e, e' \in \mathrm{supp}(\mathcal{E})$, it holds that $\mathrm{P}\left(Y \mid \psi\left(\boldsymbol{X}_e\right)\right) = \mathrm{P}\left(Y \mid \psi\left(\boldsymbol{X}_{e'}\right)\right).$*

This assumption implies that for time series $\boldsymbol{X}$ observed in different environments, invariant rationales exist and their relationship with the corresponding labels remains stable. This stability ensures that predictions remain consistent across various environments, relying on these rationales. Assuming $\psi(\cdot)$ represents the representation function $r_\theta(\cdot)$ parameterized by $\theta$, then it follows that

$$r_{\theta^*}(\boldsymbol{X}_e) = r_{\theta^*}(\boldsymbol{X}_{e'}) = \psi(\boldsymbol{X}). \tag{3}$$

In contrastive representation learning, where labels are not available, the theoretical analysis of the downstream performance is challenging. To address this, research [Zhao et al., 2022] bridges this gap by connects contrastive loss to downstream risks,

$$\mathcal{R}(h_\omega \circ r_\theta; \mathbb{P}_\pi) \leq c\|h_\omega\|\sqrt{K\sigma}(\ell_{\mathrm{align}}(r_\theta; \mathbb{P}, \pi))^{\frac{1}{4}} + \|h_\omega\|\tau(\sigma, \delta) + \sum_k \mathbb{P}_\pi(C_k)\|e_k - h_\omega \circ \mu_k(r_\theta; \mathbb{P}_\pi)\| \tag{4}$$

where $c$ is a positive constant, $\tau(\sigma, \delta)$ refers to a set of constants determined by the $(\sigma, \delta)$-augmentation, and $C_k$ corresponds to the sample set for class $k$. The first term is optimized during contrastive pre-training. The second term depends on data augmentations $(\sigma, \delta)$. The third term, related to the linear layer $h_\omega$, is optimized in downstream tasks. As shown by Eq. (4), contrastive learning on distribution $\mathbb{P}$ with augmentation function $\pi$ essentially optimizes the upper bound of the supervised risk.

Each environment $\mathcal{E}$ corresponds to a domain distribution $\mathbb{P}_{S_i}$. To learn an invariant representation over the domain set $\mathcal{P}$, we first provide a mathematical definition of invariant risk minimization.

**Definition 1** (Invariant Risk Minimization [Arjovsky et al., 2019]). *If there exists a classifier $h_0$ that is optimal for all domains in $\mathcal{P}$, i.e., $h_0 \in \mathrm{argmin}_h \mathcal{R}\left(h \circ r_\theta; \mathbb{P}_{S_i}\right), \forall \mathbb{P}_{S_i} \in \mathcal{P}$, then the representation function $r_\theta$ elicits an invariant predictor $h_0 \circ r_\theta$ across the domain set $\mathcal{P}$.*

This definition is equivalent to learning features that have a stable association with the target variable, which has been theoretically and empirically proven to improve the transferability of supervised learning across different distributions [Arjovsky et al., 2019, Zhao et al., 2022].

## 3.2 A Tri-level Learning Framework

To address OOD challenges, GroupDRO [Sagawa et al., 2019] propose a mimax formulation to minimizes the maximum domain supervised loss to enhance robustness against unseen data. According to Eq. (4), contrastive learning optimizes the upper bound of supervised risk. Thus, we extend GroupDRO by replacing the supervised loss with a self-supervised contrastive loss, aiming to learn invariant representations. We further impose constraints on the group distribution $\boldsymbol{q}$ to mitigate the risk of overfitting to specific domains. This results in a bi-level optimization problem

$$\begin{aligned} \min_{\boldsymbol{\theta}, \boldsymbol{q}} \quad & \sum_{i=1}^K q_i \ell_{\mathrm{con}}(r_\theta; \mathcal{D}_{S_i}, \pi) \\ \mathrm{s.t.} \quad & \boldsymbol{q} = \arg\max_{\boldsymbol{q}' \in \Delta^K} \sum_{i=1}^K q_i' \ell_{\mathrm{con}}(r_\theta; \mathcal{D}_{S_i}, \pi) \\ & \mathrm{s.t.}\ d(\boldsymbol{p}, \boldsymbol{q}') \leq \tau, \end{aligned} \tag{5}$$

where $d(\cdot, \cdot)$ denotes a distribution distance metric, such as KL divergence, Wasserstein distance, or Euclidean distance, $\Delta^K$ is a probability simplex, and $\tau$ is a constant. Following previous work [Qian et al., 2019], we adopt the Euclidean distance due to its strong convexity, which reults in faster convergence [Rakhlin et al., 2012]. The outer optimization seeks the best parameters across all domains to optimize overall performance, while the inner optimization, representing the group-level uncertainty, optimizes the worst-case distribution to enhance representation robustness.

**Definition 2** (Augmentation Robust Alignment Loss [Zhao et al., 2022]). *For any two augmentation methods $a, a' \in \mathcal{A}$, the robust alignment loss is defined as follows*

$$\ell_{\mathrm{ar}}(r_\theta; \mathbb{P}) := \mathbb{E}_{\boldsymbol{X} \sim \mathbb{P}} \sup_{(a, a') \in \mathcal{A}} \|r_\theta(a(\boldsymbol{X})) - r_\theta(a'(\boldsymbol{X}))\|^2. \tag{6}$$

**Theorem 1** (Upper Bound of Risk Gap Between Augmented Domains [Shen et al., 2021]). *For any two augmentation methods $a, a' \in \mathcal{A}$, representation function $r_\theta$ and classifier $h_\omega$, we have*

$$\sup_{a, a' \in \mathcal{A}} \|\mathcal{R}\left(h_\omega \circ r_\theta; \mathbb{P}_a\right) - \mathcal{R}\left(h_\omega \circ r_\theta; \mathbb{P}_{a'}\right)\| \leq c\|h_\omega\|\ell_{ar}\left(r_\theta; \mathbb{P}_{train}\right). \tag{7}$$

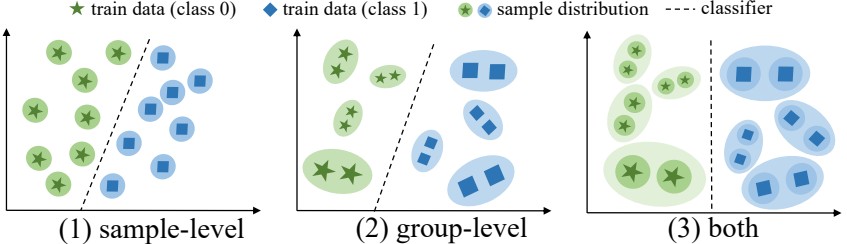

Figure 1: The depiction of sample-level, group-level, and combined uncertainties.

*Fix $r_\theta$, let $h_a = \arg\min_{h_\omega} \mathcal{R}(h_\omega \circ r; \mathbb{P}_{train})$, we have*

$$\|\mathcal{R}(h_a \circ r_\theta; \mathbb{P}_{a'}) - \mathcal{R}(h_{a'} \circ r_\theta; \mathbb{P}_{a'})\| \leq 2c(\|h_a\| + \|h_{a'}\|)\ell_{ar}(r_\theta; \mathbb{P}_{train}). \tag{8}$$

Theorem 1 states that minimizing $\ell_{ar}(r_\theta; \mathbb{P}_{train})$ makes the optimal predictor more consistent across different augmentation domains, i.e., minimize $\ell_{ar}(r_\theta; \mathbb{P}_{train})$ can enhance the invariance of the learned representation. Nonetheless, evaluating $\ell_{ar}(r; \mathbb{P}_{train})$ involves a supremum operator, and the large set $\mathcal{A}$ makes accurate computation infeasible. Therefore, we propose an approximation for $\ell_{ar}(r_\theta; \mathbb{P}_{train})$. We start with the following reasonable assumption.

**Assumption 2.** *For any pair of augmentation methods $a, a' \in \mathcal{A}$, they can be viewed as introducing specific perturbations $\boldsymbol{\delta}$ to the sample $\boldsymbol{X}$, i.e., $a(\boldsymbol{X}) = \boldsymbol{X} + \boldsymbol{\delta}_a, a'(\boldsymbol{X}) = \boldsymbol{X} + \boldsymbol{\delta}_{a'}$.*

Suppose $\boldsymbol{\delta}_a$ and $\boldsymbol{\delta}_{a'}$ are sampled from $\mathbb{P}_{perb}$, representing the distribution of perturbations induced by augmentation techniques. We adopt a Gaussian Mixture Model (GMM) [Jiao et al., 2022b] to accurately characterize the uncertain perturbation distribution. Thus, the distribution of $\boldsymbol{\delta}$ is given by

$$p(\boldsymbol{\delta}; \boldsymbol{\pi}, \boldsymbol{\mu}, \boldsymbol{\sigma}) = \sum_{m=1}^{M} \pi_m \mathcal{N}(\boldsymbol{\delta}; \mu_m, \sigma_m^2), \tag{9}$$

where $\pi_m$ represents the weight of the $m^{th}$ component in the mixture, and $\sum_{m=1}^{M} \pi_m = 1$. The expression for $\ell_{ar}(r_\theta; \mathbb{P}_{train})$ can be written as $\sup_{\boldsymbol{\delta} \sim p(\boldsymbol{\delta}; \boldsymbol{\pi}, \boldsymbol{\mu}, \boldsymbol{\sigma})} \sum_{i=1}^{K} q_i \ell_{align}(\boldsymbol{\theta}, \boldsymbol{\delta}; \mathcal{D}_{S_i})$. Consequently, we can further extend problem (5) to the following tri-level optimization problem.

$$
\begin{aligned}
\min_{\boldsymbol{\theta}, \boldsymbol{q}, \boldsymbol{\delta}} \quad & \sum_{i=1}^{K} q_i \ell_{con}(\boldsymbol{\theta}, \boldsymbol{\delta}; \mathcal{D}_{S_i}) \\
\text{s.t.} \quad & \boldsymbol{q} = \arg\max_{\boldsymbol{q}' \in \Delta^K} \sum_{i=1}^{K} q_i' \ell_{con}(\boldsymbol{\theta}, \boldsymbol{\delta}; \mathcal{D}_{S_i}) \\
& \text{s.t. } d(\boldsymbol{p}, \boldsymbol{q}') \leq \tau \\
& \boldsymbol{\delta} = \arg\max_{\boldsymbol{\delta}' \sim p(\boldsymbol{\delta}'; \boldsymbol{\pi}, \boldsymbol{\mu}, \boldsymbol{\sigma})} \sum_{i=1}^{K} q_i' \ell_{align}(\boldsymbol{\theta}, \boldsymbol{\delta}'; \mathcal{D}_{S_i}) \\
& \text{s.t. } \|\boldsymbol{\mu}\| \leq C_1, \|\boldsymbol{\sigma}\| \leq C_2, \sum_{m=1}^{M} \pi_m = 1, \pi_m \geq 0,
\end{aligned}
\tag{10}
$$

where $C_1$ and $C_2$ are constants. The third-level optimization addresses sample-level uncertainties by maximizing the alignment loss under the worst-case perturbation distribution. Figure 1 illustrates these concepts, showing how group-level and sample-level optimizations interact within the tri-level framework. To theoretically justify our approach in Eq. (10), we present the following theorem.

**Theorem 2** (Upper Bound on Target Error). *Given the previous setup, let $\mathcal{H}$ be a hypothesis space of Vapnik-Chervonenkis (VC) Dimension $d$ and $h_T^* = \min_{h \in \mathcal{H}} \epsilon_T(h)$. Let $\mathcal{P}_\alpha = \{\mathbb{P}_\alpha \mid \mathbb{P}_\alpha = \sum_i \alpha_i \mathbb{P}_{S_i}, \sum_i \alpha_i = 1, \alpha_i \geq 0\}$. If $\hat{h} \in \mathcal{H}$ is the empirical minimizer on $\mathbb{P}_\alpha$, then for any $\delta$ and $\mathbb{P}_C \in \mathcal{P}_\alpha$, with probability at least $1 - \delta$,*

$$\epsilon_T(\hat{h}) \leq 3\epsilon_T(h_T^*) + \lambda + d_{\mathcal{H}\Delta\mathcal{H}}(\mathbb{P}_C, \mathbb{P}_T) + \max_{i,j} d_{\mathcal{H}\Delta\mathcal{H}}(\mathbb{P}_{S_i}, \mathbb{P}_{S_j}) + C(\delta, m, d), \tag{11}$$

*where $\lambda = 2\sum_{i=1}^{K} \alpha_i \epsilon_{S_i}(h^*)$ and $C(\delta, m, d)$ is a statistical term. $d_{\mathcal{H}\Delta\mathcal{H}}(\cdot, \cdot)$ is a metric function which measures differences in distribution [Ben-David et al., 2010]. $\epsilon_{S_i}(h)$ and $\epsilon_T(h)$ is the source error and the target error.*

**Discussion**: Theorem 2 provides a theoretical framework for estimating performance on a new target distribution. The TTSO framework in Eq. (10) focuses on minimizing the terms $d_{\mathcal{H}\Delta\mathcal{H}}(\mathbb{P}_C, \mathbb{P}_T)$ and $\max_{i,j} d_{\mathcal{H}\Delta\mathcal{H}}(\mathbb{P}_{S_i}, \mathbb{P}_{S_j})$, thereby giving a tighter bound of target error to improve generalization ability. Proof of Theorem 2 and further discussion of our motivation are in Appendix A.2.

---

**Algorithm 1** SLA: Stratified Localization Algorithm

---

**Input**: Training datasets $\{\mathcal{D}_{S_i}\}$, learning rates $\eta_\theta, \eta_q, \eta_\delta$, number of iterations $T$.
**Output**: Optimized parameters $\boldsymbol{\theta}^*$

  1: Initialize parameters $\boldsymbol{\theta}, \boldsymbol{q}, \boldsymbol{\delta}$ and initial set of cutting planes $\mathcal{S}^0$.
  2: **for** $t = 0$ to $T - 1$ **do**
  3:     Update variable $\boldsymbol{\theta}^{(t+1)}$, $\boldsymbol{q}^{(t+1)}$ and $\boldsymbol{\delta}^{(t+1)}$ according to Eq. (19), (20) and (21)
  4:     **if** t mod k == 0 **then**
  5:       **if** $h(\boldsymbol{\theta}^{(t+1)}, \boldsymbol{q}^{(t+1)}, \boldsymbol{\delta}^{(t+1)}) > \varepsilon$ **then**
  6:         Add new cutting planes to set $\mathcal{S}^{t+1}$ according to Eq. (22)
  7:       **end if**
  8:     **end if**
  9: **end for**
10: **return** $\boldsymbol{\theta}^{(T)}$

---

However, solving the constrained tri-level optimization is extremely challenging. In the next subsection, we introduce a stratified localization algorithm to address this problem effectively.

### 3.3 Stratified Localization Algorithm

Due to the hierarchical structure of the tri-level problem, we develop a stratified version of the localization method [Boyd and Vandenberghe, 2007, Jiao et al., 2023] to tackle the problem presented in Eq. (10). First, we use exterior penalty method to reformulate the third level problem, the resulting problem is

$$
\begin{aligned}
\min_{\boldsymbol{\theta}, \boldsymbol{q}, \boldsymbol{\delta}} \quad & \sum_{i=1}^{K} q_i \ell_{\mathrm{con}}\left(\boldsymbol{\theta}, \boldsymbol{\delta}; \mathcal{D}_{S_i}\right) \\
\text{s.t.} \quad & \boldsymbol{q} = \arg\max_{\boldsymbol{q}' \in \Delta^K} \sum_{i=1}^{K} q_i' \ell_{\mathrm{con}}\left(\boldsymbol{\theta}, \boldsymbol{\delta}; \mathcal{D}_{S_i}\right) \\
& \text{s.t. } d\left(\boldsymbol{p}, \boldsymbol{q}'\right) \leq \tau \\
& \boldsymbol{\delta} = \arg\max_{\boldsymbol{\delta}' \sim p(\boldsymbol{\delta}'; \boldsymbol{\pi}, \boldsymbol{\mu}, \boldsymbol{\sigma})} \sum_{i=1}^{K} q_i' \ell_{\mathrm{align}}\left(\boldsymbol{\theta}, \boldsymbol{\delta}'; \mathcal{D}_{S_i}\right) - P_3,
\end{aligned}
\tag{12}
$$

where $P_3$ is a penalty term defined as $P_3 = \rho_1(\max(0, \|\boldsymbol{\mu}\| - C_1))^2 + \rho_2(\max(0, \|\boldsymbol{\sigma}\| - C_2))^2 + \rho_3(\sum_{m=1}^{M} \pi_m - 1)^2 + \rho_4(\max(0, -\pi_m))^2$, and $\rho_i$ are penalty coefficients.

Given that the third-level optimization is a constraint for the second-level optimization, we employ $T_3$ steps of gradient ascent to approximate the third-level problem. This technique is commonly used in previous bi-level optimization studies [Ji et al., 2021]. By defining $f_3(\boldsymbol{\theta}, \boldsymbol{q}', \boldsymbol{\delta}') = \sum_{i=1}^{K} q_i'(\ell_{\mathrm{align}}\left(\boldsymbol{\theta}, \boldsymbol{\delta}'; \mathcal{D}_{S_i}\right) - P_3$ and using the exterior penalty method, the resulting optimization problem can be expressed as

$$
\begin{aligned}
\min_{\boldsymbol{\theta}, \boldsymbol{q}, \boldsymbol{\delta}} \quad & \sum_{i=1}^{K} q_i \ell_{\mathrm{con}}\left(\boldsymbol{\theta}, \boldsymbol{\delta}; \mathcal{D}_{S_i}\right) \\
\text{s.t.} \quad & \boldsymbol{q} = \arg\max_{\boldsymbol{q}' \in \Delta^K} \sum_{i=1}^{K} q_i' \ell_{\mathrm{con}}\left(\boldsymbol{\theta}, \boldsymbol{\delta}; \mathcal{D}_{S_i}\right) - P_2,
\end{aligned}
\tag{13}
$$

where $P_2 = \lambda_1(\sum_{i=1}^{K} q_i - 1)^2 + \sum_{i=1}^{K} \lambda_2 \max(0, -q_i) + \lambda_3 \|\boldsymbol{\delta} - \boldsymbol{\delta}^{(0)} - \sum_{i=0}^{T_3 - 1} \eta_\delta \nabla_{\boldsymbol{\delta}'} f_3(\boldsymbol{\theta}, \boldsymbol{q}', \boldsymbol{\delta}')\|^2$.

Likewise, we perform $T_2$ steps of gradient ascent to replace the second level optimization problem. With the definition of $f_2(\boldsymbol{\theta}, \boldsymbol{q}', \boldsymbol{\delta}) = \sum_{i=1}^{K} q_i' \ell_{\mathrm{con}}\left(\boldsymbol{\theta}, \boldsymbol{\delta}; \mathcal{D}_{S_i}^{tr}\right) - P_2, \varphi(\boldsymbol{\theta}, \boldsymbol{\delta}) = \arg\max_{\boldsymbol{q}'} f_2(\boldsymbol{\theta}, \boldsymbol{q}', \boldsymbol{\delta})$ and $h(\boldsymbol{\theta}, \boldsymbol{q}, \boldsymbol{\delta}) = \|\boldsymbol{q} - \varphi(\boldsymbol{\theta}, \boldsymbol{\delta})\|$, Eq. (13) can be reformulated as follows

$$
\begin{aligned}
\min_{\boldsymbol{\theta}, \boldsymbol{q}, \boldsymbol{\delta}} \quad & \sum_{i=1}^{K} q_i \ell_{\mathrm{con}}\left(\boldsymbol{\theta}, \boldsymbol{\delta}; \mathcal{D}_{S_i}\right) \\
\text{s.t.} \quad & h(\boldsymbol{\theta}, \boldsymbol{q}, \boldsymbol{\delta}) = 0.
\end{aligned}
\tag{14}
$$

Let $f_1\left(\boldsymbol{\theta}, \boldsymbol{q}, \boldsymbol{\delta}\right) = \sum_{i=1}^{K} q_i \ell_{\mathrm{align}}\left(\boldsymbol{\theta}, \boldsymbol{\delta}; \mathcal{D}_{S_i}\right)$. Considering the approximations of $\boldsymbol{q}$ and $\boldsymbol{\delta}$, the above problem can be relaxed as

$$
\begin{aligned}
\min_{\boldsymbol{\theta}, \boldsymbol{q}, \boldsymbol{\delta}} \quad & f_1\left(\boldsymbol{\theta}, \boldsymbol{q}, \boldsymbol{\delta}\right) \\
\text{s.t.} \quad & h(\boldsymbol{\theta}, \boldsymbol{q}, \boldsymbol{\delta}) \leq \varepsilon,
\end{aligned}
\tag{15}
$$

where $\varepsilon > 0$ is a constant. Inspired by the polyhedral approximation method [Bürger et al., 2013], we utilize cutting planes to approximate the feasible region with respect to $h(\boldsymbol{\theta}, \boldsymbol{q}, \boldsymbol{\delta}) \leq \varepsilon$. In the

$(t + 1)^{th}$ iteration, the set of cutting planes, denoted as $\mathcal{S}^t$, is defined as follows

$$\mathcal{S}^t = \{\boldsymbol{a}_i^\top \boldsymbol{\theta} + \boldsymbol{b}_i^\top \boldsymbol{q} + \boldsymbol{c}_i^\top \boldsymbol{\delta} + d_i \leq 0, i = 1, \cdots, |\mathcal{S}^t|\}, \tag{16}$$

where $\boldsymbol{a}_i \in \mathbb{R}^N, \boldsymbol{b}_i \in \mathbb{R}^M, \boldsymbol{c}_i \in \mathbb{R}^H, d_i \in \mathbb{R}^1$, and $|\mathcal{S}^t|$ represents the number of cutting planes in $\mathcal{S}^t$. Then Eq. (15) can be expressed as the following approximation problem

$$\begin{aligned} \min_{\boldsymbol{\theta}, \boldsymbol{q}, \boldsymbol{\delta}} \quad & f_1(\boldsymbol{\theta}, \boldsymbol{q}, \boldsymbol{\delta}) \\ \text{s.t.} \quad & \boldsymbol{a}_i^\top \boldsymbol{\theta} + \boldsymbol{b}_i^\top \boldsymbol{q} + \boldsymbol{c}_i^\top \boldsymbol{\delta} + d_i \leq 0, \quad i = 1, \cdots, |\mathcal{S}^t|. \end{aligned} \tag{17}$$

The penalty function with respect to Eq. (17) can be described as

$$F(\boldsymbol{\theta}, \boldsymbol{q}, \boldsymbol{\delta}) = f_1(\boldsymbol{\theta}, \boldsymbol{q}, \boldsymbol{\delta}) + \sum_i \lambda_i \max(0, \boldsymbol{a}_i^\top \boldsymbol{\theta} + \boldsymbol{b}_i^\top \boldsymbol{q} + \boldsymbol{c}_i^\top \boldsymbol{\delta} + d_i)^2. \tag{18}$$

In $(t + 1)^{th}$ iteration, the variables are updated as follows

$$\boldsymbol{\theta}^{t+1} = \boldsymbol{\theta}^t - \eta_\theta \nabla_\theta F(\boldsymbol{\theta}^t, \boldsymbol{q}^t, \boldsymbol{\delta}^t), \tag{19}$$

$$\boldsymbol{q}^{t+1} = \boldsymbol{q}^t - \eta_q \nabla_q F(\boldsymbol{\theta}^t, \boldsymbol{q}^t, \boldsymbol{\delta}^t), \tag{20}$$

$$\boldsymbol{\delta}^{t+1} = \boldsymbol{\delta}^t - \eta_\delta \nabla_\delta F(\boldsymbol{\theta}^t, \boldsymbol{q}^t, \boldsymbol{\delta}^t). \tag{21}$$

Throughout the iteration process, the set of cutting planes $\mathcal{S}^t$ is updated every $k$ iterations for a tighter and more accurate polyhedral approximation. Before adding new cutting planes, we first check whether $(\boldsymbol{\theta}^{t+1}, \boldsymbol{q}^{t+1}, \boldsymbol{\delta}^{t+1})$ is a solution for Eq. (15). If it is not a feasible solution to Eq. (15), i.e., $h(\boldsymbol{\theta}, \boldsymbol{q}, \boldsymbol{\delta}) > \varepsilon$, new cutting planes are added to $\mathcal{S}^t$ based on Theorem 3 and Proposition 22. Algorithm 1 provides details of the proposed method.

**Theorem 3.** *Let $T_2 = 1$. If a first-order Taylor expansion is applied to the function $f_2(\boldsymbol{\theta}, \boldsymbol{q}, \boldsymbol{\delta})$ at the point $(\overline{\boldsymbol{\theta}}, \overline{\boldsymbol{\delta}})$, it follows that the function $h(\boldsymbol{\theta}, \boldsymbol{q}, \boldsymbol{\delta})$ is convex with respect to $(\boldsymbol{\theta}, \boldsymbol{q}, \boldsymbol{\delta})$. The detailed proof can be found in Appendix A.3.*

**Proposition 1.** *Given the convexity of the function $h(\boldsymbol{\theta}, \boldsymbol{q}, \boldsymbol{\delta})$, a new cutting plane is generated when the condition $h(\boldsymbol{\theta}, \boldsymbol{q}, \boldsymbol{\delta}) > \varepsilon$ is not met. This cutting plane is formally expressed as*

$$h(\boldsymbol{\theta}^{t+1}, \boldsymbol{q}^{t+1}, \boldsymbol{\delta}^{t+1}) + \begin{bmatrix} \nabla_\theta h(\boldsymbol{\theta}^{t+1}, \boldsymbol{q}^{t+1}, \boldsymbol{\delta}^{t+1}) \\ \nabla_q h(\boldsymbol{\theta}^{t+1}, \boldsymbol{q}^{t+1}, \boldsymbol{\delta}^{t+1}) \\ \nabla_\delta h(\boldsymbol{\theta}^{t+1}, \boldsymbol{q}^{t+1}, \boldsymbol{\delta}^{t+1}) \end{bmatrix}^\top \begin{bmatrix} \boldsymbol{\theta} - \boldsymbol{\theta}^{t+1} \\ \boldsymbol{q} - \boldsymbol{q}^{t+1} \\ \boldsymbol{\delta} - \boldsymbol{\delta}^{t+1} \end{bmatrix} \leq \varepsilon. \tag{22}$$

For the detailed derivation and proof of Proposition 22, please see Appendix A.1.

### 3.4 TTSO for Fine-tuning LLMs

LLMs have garnered considerable attention in time series applications [Jin et al., 2023, Zhou et al., 2023]. The emergent abilities of LLMs, especially in OOD scenarios, largely depends on the robustness of their representations[Wang et al., 2023, Chu et al., 2023]. This section connects the established theoretical foundation with the practical application of fine-tuning LLMs for time series OOD generalization. We adapt TTSO framework for fine-tuning LLMs to enhance the performance in time series OOD generalization. Our proposed method involves a dual-stage fine-tuning method tailored for time series. The main process of fine-tuning are described below.

**Time Series Pre-processing.** Preprocessing starts with an input projection layer to bridge the gap in dimensions between raw time series data and the LLM's native embedding dimension. This step is crucial for the LLM's effective integration of time series. Following this, positional encoding is applied to preserve the sequential integrity of the time series.

**Dual-stage Fine-tuning Method.** In the first stage, we employ TTSO framework to fine-tune LLMs, in line with the previously mentioned tri-level optimization framework as illustrate in Eq. (10). We adopt the contrastive loss function designed for time series from Yue et al. [2022]. In the second stage, the learned weights of the LLM, including the projection layer, are transferred to the downstream fine-tuning stage for time series classification. To retain the knowledge learned by the LLM from the corpus, we follow Chang et al. [2023], Zhou et al. [2023] by fixing the weights of the fully connected and attention layers, using Layer Normalization Tuning [Lu et al., 2022a] to adjust only the layer normalization parameters, making the affine transformation trainable.

**Constrained Optimization for Fine-Tuning.** Research [Wortsman et al., 2022] indicates that adopting radical strategies for fine-tuning models, such as larger learning rates, can reduce out-of-

distribution robustness. Unconstrained optimization of model parameters during fine-tuning can lead to knowledge forgetting issues and decrease the model's generalization ability, as mentioned in Xuhong et al. [2018]. Therefore, during fine-tuning for downstream tasks, we impose constraints on the parameters, following Xuhong et al. [2018], resulting in the following optimization problem

$$\begin{aligned} \min_{\boldsymbol{\theta}} \quad & \ell_{cls}\left(r_{\theta} \circ h_{\omega}; \mathcal{D}\right) \\ \text{s.t.} \quad & \|\boldsymbol{\theta} - \boldsymbol{\theta}_0\| \le \gamma, \end{aligned} \tag{23}$$

where $\boldsymbol{\theta}_0$ and $\boldsymbol{\theta}$ respectively denote the weights from the first and second fine-tuning phases of the LLMs. More details of fine-tuning LLMs can be found in appendix D.

## 4 Convergence Analysis

**Assumption 3** (**Lipschitz Continuity of Gradient**). *Assume that the gradient of the function $F$ is $L$-Lipschitz continuous gradient, i.e., for any $\boldsymbol{x}, \boldsymbol{y}$, there exists $L > 0$ such that:*

$$\|\nabla F(\boldsymbol{x}) - \nabla F(\boldsymbol{y})\| \le L \|\boldsymbol{x} - \boldsymbol{y}\|. \tag{24}$$

**Assumption 4** (**Unbiasedness and Variance Bound of Stochastic Gradients**). *Assume for the stochastic gradients $g_\theta, g_q, g_\delta$, the following conditions are satisfied*

$$\begin{aligned} \mathbb{E}_{\zeta_j^t}[g_\theta(\boldsymbol{\theta}^t, \boldsymbol{q}^t, \boldsymbol{\delta}^t; \zeta_j^t) - \nabla_\theta F(\boldsymbol{\theta}^t, \boldsymbol{q}^t, \boldsymbol{\delta}^t)] &= 0, \\ \mathbb{E}_{\zeta_j^t}[g_q(\boldsymbol{\theta}^t, \boldsymbol{q}^t, \boldsymbol{\delta}^t; \zeta_j^t) - \nabla_q F(\boldsymbol{\theta}^t, \boldsymbol{q}^t, \boldsymbol{\delta}^t)] &= 0, \\ \mathbb{E}_{\zeta_j^t}[g_\delta(\boldsymbol{\theta}^t, \boldsymbol{q}^t, \boldsymbol{\delta}^t; \zeta_j^t) - \nabla_\delta F(\boldsymbol{\theta}^t, \boldsymbol{q}^t, \boldsymbol{\delta}^t)] &= 0, \\ \mathbb{E}_{\zeta_j^t}[\|g_\theta(\boldsymbol{\theta}^t, \boldsymbol{q}^t, \boldsymbol{\delta}^t; \zeta_j^t) - \nabla_\theta F(\boldsymbol{\theta}^t, \boldsymbol{q}^t, \boldsymbol{\delta}^t)\|^2] &\le \sigma_1^2, \\ \mathbb{E}_{\zeta_j^t}[\|g_q(\boldsymbol{\theta}^t, \boldsymbol{q}^t, \boldsymbol{\delta}^t; \zeta_j^t) - \nabla_q F(\boldsymbol{\theta}^t, \boldsymbol{q}^t, \boldsymbol{\delta}^t)\|^2] &\le \sigma_2^2, \\ \mathbb{E}_{\zeta_j^t}[\|g_\delta(\boldsymbol{\theta}^t, \boldsymbol{q}^t, \boldsymbol{\delta}^t; \zeta_j^t) - \nabla_\delta F(\boldsymbol{\theta}^t, \boldsymbol{q}^t, \boldsymbol{\delta}^t)\|^2] &\le \sigma_3^2, \end{aligned} \tag{25}$$

*where $\mathbb{E}_{\zeta_j^t}[\cdot]$ denotes the expectation over the $\zeta_j^t$.*

**Assumption 5** (**Bounded Gradient**). *Assume that the gradient of the function $F$ is bounded, i.e., $\forall t, \|\nabla_\theta F(\boldsymbol{\theta}^t, \boldsymbol{q}^t, \boldsymbol{\delta}^t)\|^2 \le \alpha_1^2, \|\nabla_q F(\boldsymbol{\theta}^t, \boldsymbol{q}^t, \boldsymbol{\delta}^t)\|^2 \le \alpha_2^2, \|\nabla_\delta F(\boldsymbol{\theta}^t, \boldsymbol{q}^t, \boldsymbol{\delta}^t)\|^2 \le \alpha_3^2.$*

**Definition 3** ($\epsilon$-**Stationary Point**). *Following Xu et al. [2023], Jiao et al. [2024], a point $(\boldsymbol{\theta}, \boldsymbol{q}, \boldsymbol{\delta})$ is considered an $\epsilon$-stationary point (where $\epsilon > 0$) of a differentiable function $F$ if the sum of squares of its gradients on these variables satisfies $\|\nabla G^t\| \le \epsilon$. Let $T(\epsilon)$ be the index of the first iteration that satisfies $\|\nabla G^t\| \le \epsilon$, i.e., $T(\epsilon) = \min\{t \mid \|\nabla G^t\| \le \epsilon, t > t_1\}$.*

**Theorem 4** (**Convergence Guarantee**). *With the continuous addition of cutting planes, the optimal objective value of the approximated problem, delineated in Eq. (17), is guaranteed to converge monotonically. For further details, see the proof of Theorem 4 in appendix A.4.*

**Theorem 5** (**Convergence Rate**). *Under the assumptions 3, 4, and 5, by setting the step-sizes as $\eta_\theta = \eta_q = \eta_\delta = \frac{1}{\sqrt{T_1 - t_1}}$ and the batch size as $B$, for a given $\epsilon$, it follows that*

$$T(\epsilon) \sim \mathcal{O}\left(t_1 + \frac{L^2(m(\alpha_1^2 + \alpha_2^2 + \alpha_3^2) + \sigma_1^2 + \sigma_2^2 + \sigma_3^2)^2}{4m^2(\epsilon - F(\boldsymbol{\theta}^{T_1}, \boldsymbol{q}^{T_1}, \boldsymbol{\delta}^{T_1}) + F^*)^2}\right), \tag{26}$$

*where $F^*$ represents the lower bound of $F$. The proof of Theorem 5 is detailed in appendix A.5.*

## 5 Experiment

To evaluate the proposed TTSO framework, we conduct experiments on 6 real-world time series datasets using the leave-one-domain-out setting, including HHAR [Blunck et al., 2015], PAMAP [Reiss, 2012], WESAD [Philip Schmidt et al., 2018], SWELL [Koldijk et al., 2014], USC-HAD[Zhang and Sawchuk, 2012] and DSADS [Barshan and Altun, 2013]. We compare with baseline method ERM [Vapnik, 1991] and 8 general OOD generalization methods: IRM [Arjovsky et al., 2019], GroupDRO [Sagawa et al., 2019], ANDMask [Parascandolo et al., 2020], RSC [Huang et al., 2020], Mixup [Zhang et al., 2017], VERx [Krueger et al., 2021], DIFEX[Lu et al., 2022b]. And we further compare with 2 recent strong approach in time series: AdaRNN[Du et al., 2021] and GILE [Qian et al., 2021]. We also include DIVERSIFY[Lu et al., 2023], DFDG[Zhang et al., 2021], and CCDG[Ragab et al., 2022], three methods specifically designed for time series OOD

generalization. To guarantee a fair comparison, we implement all methods using the same backbone architecture (except AdaRNN and GILE), TCN [Bai et al., 2018], a model widely used in time series analysis. In addition, we fine-tune the pre-trained Large Language Model, GPT2[Radford et al., 2018], within our TTSO framework to harness its sophisticated representation learning capabilities. Detailed information regarding datasets, domain setting, data pre-processing, network architecture and hyperparameters are provided in Appendix B.1, B.2, B.3 and C.

## 5.1 Main Results

We report the average results over 3 runs for each dataset, along with the standard deviation. The results for the HHAR, PAMAP, and WESAD datasets are shown in Tables 1, where our method outperforms the second-best baseline by 2.8%, 4.8%, and 4.9% respectively. Additional results are provided in Appendix E.1 (Table 4). These results demonstrate the superiority and effectiveness of the TTSO framework, as it accounts for both sample-level and group-level uncertainties, which optimizes the upper bound in Theorem 2.

Compared to traditional methods like ERM, IRM, and GroupDRO, both TTSO and TTSO* show more consistent and generally superior performance in OOD generalization, highlighting the advantages of a tri-level learning framework. The TTSO* method, which incorporates LLM fine-tuning, consistently outperforms other approaches, demonstrating the effectiveness of LLM with TTSO fine-tuning in enhancing OOD generalization for time series. Concurrently, the TTSO method, even without LLM fine-tuning, shows strong generalization performance, especially on the HHAR dataset where it closely matches TTSO* results. This indicates that the TTSO framework is highly effective in generalizing across different scenarios, even in the absence of LLM.

Table 1: Classification accuracy (%) on HHAR, PAMAP, and WESAD datasets. **Bold** indicates the best, underline the second-best performance. Standard deviation is shown in the lower right corner.

| Method | HHAR | | | | | PAMAP | | | | | WESAD | | | | | ALL |
| | A | B | C | D | AVG | A | B | C | D | AVG | A | B | C | D | AVG | AVG |
|---|---|---|---|---|---|---|---|---|---|---|---|---|---|---|---|---|
| ADARNN | $73.0_{0.01}$ | $65.0_{0.04}$ | $76.8_{0.02}$ | $67.0_{0.00}$ | 70.5 | $71.8_{0.01}$ | $72.4_{0.01}$ | $53.8_{0.01}$ | $74.2_{0.04}$ | 68.0 | $40.8_{0.02}$ | $72.2_{0.00}$ | $63.4_{0.02}$ | $47.8_{0.03}$ | 56.0 | 64.8 |
| GILE | $65.3_{0.01}$ | $61.5_{0.01}$ | $79.2_{0.00}$ | $58.4_{0.03}$ | 66.1 | $70.1_{0.02}$ | $74.5_{0.00}$ | $45.6_{0.02}$ | $66.0_{0.01}$ | 64.1 | $42.2_{0.01}$ | $72.7_{0.02}$ | $70.5_{0.00}$ | $46.9_{0.02}$ | 58.1 | 62.8 |
| ERM | $71.6_{0.01}$ | $66.4_{0.01}$ | $78.3_{0.02}$ | $68.6_{0.01}$ | 71.2 | $72.1_{0.01}$ | $81.8_{0.00}$ | $58.9_{0.00}$ | $68.7_{0.01}$ | 70.4 | $44.5_{0.02}$ | $71.4_{0.01}$ | $65.8_{0.02}$ | $48.0_{0.01}$ | 57.5 | 66.4 |
| IRM | $72.8_{0.05}$ | $63.9_{0.01}$ | $79.2_{0.01}$ | $68.1_{0.00}$ | 71.0 | $71.4_{0.02}$ | $83.1_{0.02}$ | $58.8_{0.04}$ | $71.1_{0.02}$ | 71.1 | $45.1_{0.01}$ | $71.8_{0.01}$ | $67.0_{0.02}$ | $45.5_{0.01}$ | 57.4 | 66.5 |
| GroupDRO | $71.1_{0.01}$ | $66.1_{0.01}$ | $75.2_{0.01}$ | $67.5_{0.02}$ | 70.0 | $71.3_{0.01}$ | $80.7_{0.00}$ | $57.9_{0.01}$ | $70.3_{0.02}$ | 70.0 | $50.7_{0.01}$ | $70.2_{0.02}$ | $62.3_{0.00}$ | $53.7_{0.02}$ | 59.2 | 66.4 |
| ANDMask | $73.0_{0.03}$ | $62.3_{0.01}$ | $\mathbf{81.4}_{0.03}$ | $68.9_{0.00}$ | 71.4 | $74.0_{0.01}$ | $81.0_{0.03}$ | $55.3_{0.02}$ | $72.3_{0.00}$ | 70.6 | $44.2_{0.02}$ | $70.0_{0.03}$ | $61.7_{0.00}$ | $47.6_{0.02}$ | 55.9 | 66.0 |
| RSC | $76.3_{0.05}$ | $63.9_{0.01}$ | $78.4_{0.05}$ | $64.3_{0.01}$ | 70.7 | $74.6_{0.02}$ | $84.3_{0.01}$ | $58.9_{0.00}$ | $72.5_{0.01}$ | 72.6 | $56.9_{0.02}$ | $70.8_{0.03}$ | $67.2_{0.01}$ | $57.4_{0.02}$ | 63.1 | 68.8 |
| Mixup | $73.1_{0.02}$ | $65.9_{0.01}$ | $79.1_{0.02}$ | $69.5_{0.00}$ | 71.9 | $73.6_{0.01}$ | $\underline{87.3}_{0.01}$ | $59.3_{0.00}$ | $69.9_{0.01}$ | 72.5 | $52.9_{0.01}$ | $70.0_{0.02}$ | $73.6_{0.02}$ | $\underline{64.4}_{0.01}$ | 65.2 | 69.9 |
| VERx | $67.5_{0.01}$ | $63.2_{0.02}$ | $\underline{81.2}_{0.03}$ | $68.0_{0.00}$ | 70.0 | $74.2_{0.00}$ | $85.5_{0.00}$ | $59.5_{0.00}$ | $70.0_{0.00}$ | 72.3 | $58.3_{0.02}$ | $72.0_{0.00}$ | $66.3_{0.02}$ | $50.5_{0.02}$ | 61.8 | 68.0 |
| DIFEX | $71.5_{0.01}$ | $62.0_{0.01}$ | $81.5_{0.02}$ | $65.5_{0.01}$ | 70.1 | $73.6_{0.01}$ | $84.1_{0.01}$ | $59.2_{0.01}$ | $71.4_{0.01}$ | 72.1 | $48.2_{0.01}$ | $71.7_{0.02}$ | $63.2_{0.01}$ | $51.3_{0.01}$ | 58.6 | 66.9 |
| DFDG | $71.2_{0.01}$ | $65.8_{0.00}$ | $74.1_{0.02}$ | $\underline{70.4}_{0.00}$ | 70.3 | $73.1_{0.01}$ | $80.5_{0.03}$ | $59.2_{0.02}$ | $70.2_{0.01}$ | 70.8 | $49.8_{0.03}$ | $71.6_{0.01}$ | $71.1_{0.00}$ | $50.7_{0.02}$ | 60.8 | 67.3 |
| CCDG | $73.0_{0.02}$ | $63.2_{0.00}$ | $77.3_{0.00}$ | $\mathbf{72.4}_{0.00}$ | 71.5 | $72.3_{0.01}$ | $84.8_{0.00}$ | $56.6_{0.01}$ | $72.1_{0.00}$ | 71.5 | $54.5_{0.03}$ | $70.5_{0.01}$ | $69.8_{0.03}$ | $54.1_{0.02}$ | 62.2 | 68.4 |
| DIVERSIFY | $73.7_{0.01}$ | $64.2_{0.01}$ | $78.9_{0.01}$ | $71.2_{0.01}$ | 71.8 | $74.0_{0.02}$ | $84.0_{0.03}$ | $56.5_{0.00}$ | $\underline{72.9}_{0.03}$ | 72.0 | $57.6_{0.02}$ | $\mathbf{73.0}_{0.00}$ | $72.6_{0.01}$ | $57.1_{0.02}$ | 64.6 | 69.3 |
| TTSO | $\underline{76.6}_{0.00}$ | $\mathbf{67.5}_{0.00}$ | $80.2_{0.01}$ | $68.1_{0.01}$ | $\underline{73.1}$ | $75.3_{0.01}$ | $86.1_{0.01}$ | $\underline{60.5}_{0.01}$ | $72.5_{0.01}$ | $\underline{73.6}$ | $\mathbf{59.9}_{0.01}$ | $71.3_{0.01}$ | $76.2_{0.00}$ | $63.0_{0.01}$ | $\underline{67.6}$ | $\underline{71.4}$ |
| TTSO* | $\mathbf{77.6}_{0.02}$ | $\underline{67.3}_{0.01}$ | $80.6_{0.00}$ | $69.9_{0.01}$ | **73.9** | $78.5_{0.02}$ | $\mathbf{89.6}_{0.00}$ | $\mathbf{61.4}_{0.01}$ | $\mathbf{75.0}_{0.01}$ | **76.1** | $\underline{59.5}_{0.00}$ | $\underline{71.9}_{0.03}$ | $\mathbf{77.3}_{0.01}$ | $\mathbf{65.0}_{0.00}$ | **68.4** | **72.8** |

## 5.2 Ablation Study

This ablation study is conducted to further understand the impact of our TTSO framework's fine-tuning on model performance. We compare four distinct variants: a pretrained GPT2 fine-tuned with TTSO (TTSO$^{++}$), a pretrained GPT2 without TTSO fine-tuning (TTSO$^{+-}$), a randomly initialized GPT2 fine-tuned with TTSO (TTSO$^{-+}$), and a randomly initialized GPT2 without TTSO fine-tuning (TTSO$^{--}$). This comparison helps in quantifying the effectiveness of the TTSO fine-tuning strategy in enhancing the model's OOD generalization capabilities.

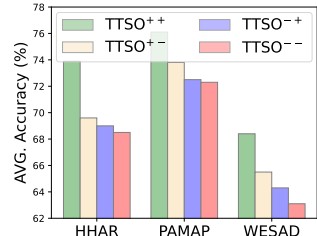

Figure 2: Ablation study of TTSO*

The ablation results are presented in Figure 2. From this results, we can see that: (a) TTSO$^{++}$ demonstrates the best performance in all scenarios, further validating that the combination of a pre-trained GPT2 model with TTSO fine-tuning can significantly improve the model's OOD generalization capabilities. (b) Although TTSO$^{+-}$ does not employ TTSO fine-tuning, it still exhibits relatively good performance. This suggests that the pre-trained GPT2 model has an intrinsic capacity for OOD generalization, consistent with previous empirical studies [Zheng et al., 2022, Hendrycks et al., 2020]. (c) Compared to TTSO$^{--}$, TTSO$^{-+}$ applies TTSO to fine-tune on a randomly initialized GPT2, TTSO$^{++}$ achieves improved performance.

This demonstrates that even in the absence of a pre-trained model, TTSO fine-tuning can effectively enhance the model's OOD generalization capabilities, though not as significantly as that with a pre-trained GPT2.

## 6 Conclusion

Existing OOD generalization methods mainly focus on sample-level uncertainties or group-level uncertainties, often overlooking the interplay between these two aspects. In light of this, we propose the TTSO framework to integrate both sample-level and group-level uncertainties within a unified tri-level learning approach, thereby enhancing the model's robustness and adaptability in facing diverse and unforeseen distribution shifts. In addition, this innovative framework introduces a fresh perspective for the development and analysis of the Out-of-Distribution (OOD) generalization problem. Based on this formulation, we develop a stratified localization algorithm for the tri-level optimization problem and provide theoretical analysis regarding the iteration complexity of the proposed algorithm. Comprehensive studies have been carried out to assess the performance of the proposed algorithm and substantiate the theoretic claims. It is seen that TTSO with LLM can considerably improves the performance of time series OOD generalization.

## 7 Acknowledgements

This work was supported in part by the National Natural Science Foundation of China under Grant 12371519 and 61771013; in part by Asiainfo Technologies; in part by the Fundamental Research Funds for the Central Universities of China; and in part by the Fundamental Research Funds of Shanghai Jiading District.

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

# Appendix

# A   Theoretical Proofs and Discussion

## A.1   Proof of Proposition 22

Since $h(\boldsymbol{\theta}, \boldsymbol{q}, \boldsymbol{\delta})$ is a convex function, we have that

$$h(\boldsymbol{\theta}, \boldsymbol{q}, \boldsymbol{\delta}) \geq h(\boldsymbol{\theta}^{t+1}, \boldsymbol{q}^{t+1}, \boldsymbol{\delta}^{t+1}) + \begin{bmatrix} \nabla_\theta h(\boldsymbol{\theta}^{t+1}, \boldsymbol{q}^{t+1}, \boldsymbol{\delta}^{t+1}) \\ \nabla_q h(\boldsymbol{\theta}^{t+1}, \boldsymbol{q}^{t+1}, \boldsymbol{\delta}^{t+1}) \\ \nabla_\delta h(\boldsymbol{\theta}^{t+1}, \boldsymbol{q}^{t+1}, \boldsymbol{\delta}^{t+1}) \end{bmatrix}^\top \begin{bmatrix} \boldsymbol{\theta} - \boldsymbol{\theta}^{t+1} \\ \boldsymbol{q} - \boldsymbol{q}^{t+1} \\ \boldsymbol{\delta} - \boldsymbol{\delta}^{t+1} \end{bmatrix}. \tag{27}$$

According to theorem 3, combine with Eq. (15) and Eq. (27), the new cutting plane will be generated as

$$h(\boldsymbol{\theta}^{t+1}, \boldsymbol{q}^{t+1}, \boldsymbol{\delta}^{t+1}) + \begin{bmatrix} \nabla_\theta h(\boldsymbol{\theta}^{t+1}, \boldsymbol{q}^{t+1}, \boldsymbol{\delta}^{t+1}) \\ \nabla_q h(\boldsymbol{\theta}^{t+1}, \boldsymbol{q}^{t+1}, \boldsymbol{\delta}^{t+1}) \\ \nabla_\delta h(\boldsymbol{\theta}^{t+1}, \boldsymbol{q}^{t+1}, \boldsymbol{\delta}^{t+1}) \end{bmatrix}^\top \begin{bmatrix} \boldsymbol{\theta} - \boldsymbol{\theta}^{t+1} \\ \boldsymbol{q} - \boldsymbol{q}^{t+1} \\ \boldsymbol{\delta} - \boldsymbol{\delta}^{t+1} \end{bmatrix} \leq \varepsilon. \tag{28}$$

From the inequalities, we can derive the coefficients $\boldsymbol{a}_i, \boldsymbol{b}_i, \boldsymbol{c}_i$ and $d_i$ as follows

$$\begin{aligned} \boldsymbol{a}_i &= \nabla_\theta h(\boldsymbol{\theta}^{t+1}, \boldsymbol{q}^{t+1}, \boldsymbol{\delta}^{t+1}), \\ \boldsymbol{b}_i &= \nabla_q h(\boldsymbol{\theta}^{t+1}, \boldsymbol{q}^{t+1}, \boldsymbol{\delta}^{t+1}), \\ \boldsymbol{c}_i &= \nabla_\delta h(\boldsymbol{\theta}^{t+1}, \boldsymbol{q}^{t+1}, \boldsymbol{\delta}^{t+1}), \\ d_i &= h(\boldsymbol{\theta}^{t+1}, \boldsymbol{q}^{t+1}, \boldsymbol{\delta}^{t+1}) - \nabla_\theta h(\boldsymbol{\theta}^{t+1}, \boldsymbol{q}^{t+1}, \boldsymbol{\delta}^{t+1})^\top \boldsymbol{\theta}^{t+1} \\ &\quad - \nabla_q h(\boldsymbol{\theta}^{t+1}, \boldsymbol{q}^{t+1}, \boldsymbol{\delta}^{t+1})^\top \boldsymbol{q}^{t+1} - \nabla_\delta h(\boldsymbol{\theta}^{t+1}, \boldsymbol{q}^{t+1}, \boldsymbol{\delta}^{t+1})^\top \boldsymbol{\delta}^{t+1} - \varepsilon, \end{aligned} \tag{29}$$

which concludes the proof.

## A.2   Proof of Theorem 2

### A.2.1   Background

For a sample distribution $\mathbb{P}_S$ on inputs space $\mathcal{X}$ with a binary labeling function $f$ and a hypothesis $h$, the error (or risk) is defined as follows

$$\epsilon_S(h, f) = \mathrm{E}_{\boldsymbol{x} \sim \mathbb{P}_S}[|h(\boldsymbol{x}) - f(\boldsymbol{x})|]. \tag{30}$$

For simplicity, we use the shorthand $\epsilon_S(h) = \epsilon_S(h, f_S)$ for source risk and $\epsilon_T(h) = \epsilon_T(h, f_D)$ for target risk, where $f_S$ and $f_D$ represent the labeling function of source and target domain, respectively.

To bound the target error, following Ben-David et al. [2010], we define the $\mathcal{H}$-divergence. Given source distribution $\mathbb{P}_S$ and target distribution $\mathbb{P}_T$ over input space $\mathcal{X}$, let $\mathcal{H}$ be a hypothesis class on $\mathcal{X}$. The $\mathcal{H}$-divergence between $\mathbb{P}_S$ and $\mathbb{P}_T$ is

$$d_\mathcal{H}(\mathbb{P}_S, \mathbb{P}_T) = 2 \sup_{h \in \mathcal{H}} |\mathrm{Pr}_{\mathbb{P}_S}[I(h)] - \mathrm{Pr}_{\mathbb{P}_T}[I(h)]|, \tag{31}$$

where $I_h = \{\boldsymbol{x} \in \mathcal{X} : h(x) = 1, h \in \mathcal{H}\}$. In addition, for a hypothesis space $\mathcal{H}$, the symmetric difference hypothesis space $d_{\mathcal{H}\Delta\mathcal{H}}$ is the set of hypotheses. And the $d_{\mathcal{H}\Delta\mathcal{H}}$ in [Ben-David et al., 2010] is defined as

$$d_{\mathcal{H}\Delta\mathcal{H}}(\mathbb{P}_S, \mathbb{P}_T) = 2 \sup_{h,h' \in \mathcal{H}} |\mathrm{Pr}_{\boldsymbol{x} \sim \mathbb{P}_S}[h(\boldsymbol{x}) \neq h'(\boldsymbol{x})] - \mathrm{Pr}_{\boldsymbol{x} \sim \mathbb{P}_T}[h(\boldsymbol{x}) \neq h'(\boldsymbol{x})]|. \tag{32}$$

**Theorem 6.** *(Modified from Theorem 4 in [Ben-David et al., 2010]) Let $\mathcal{H}$ be a hypothesis space of VC dimension $d$. For each $i \in \{1, \ldots, K\}$, let $\mathcal{D}_{S_i}$ be a labeled sample of size $\beta_i m$ drawn from $\mathbb{P}_{S_i}$ and labeled according to function $f_{S_i}$. If $\hat{h} \in \mathcal{H}$ is the empirical minimizer of $\hat{\epsilon}_\alpha(h)$ for a fixed weight vector $\boldsymbol{\alpha}$ on these samples and $h_T^* = \min_{h \in \mathcal{H}} \epsilon_T(h)$ is the target error minimizer, then for any $\delta \in (0, 1)$, with probability at least $1 - \delta$,*

$$\epsilon_T(\hat{h}) \leq \epsilon_T(h_T^*) + \sum_{i=1}^N \alpha_i \left(2\lambda_i + d_{\mathcal{H}\Delta\mathcal{H}}(\mathbb{P}_{S_i}, \mathbb{P}_T)\right) + 2\sqrt{\left(\sum_{i=1}^N \frac{\alpha_i^2}{\beta_i}\right)\left(\frac{d\log(2m) - \log(\delta)}{2m}\right)}, \tag{33}$$

*where $\lambda_i = \min_{h \in \mathcal{H}} \{\epsilon_T(h) + \epsilon_{S_i}(h)\}$.*

### A.2.2 Proof

According to the definition of $d_{\mathcal{H}\Delta\mathcal{H}}$ in Eq. (32), we have

$$
\begin{aligned}
d_{\mathcal{H}\Delta\mathcal{H}}\left(\mathbb{P}_{S_i}, \mathbb{P}_T\right) &= 2 \sup_{h,h'\in\mathcal{H}} \left|\Pr_{\boldsymbol{x}\sim\mathbb{P}_{S_i}}\left[h(\boldsymbol{x})\neq h'(\boldsymbol{x})\right] - \Pr_{\boldsymbol{x}\sim\mathbb{P}_T}\left[h(\boldsymbol{x})\neq h'(\boldsymbol{x})\right]\right| \\
&= 2 \sup_{h,h'\in\mathcal{H}} \left|\left(\Pr_{\boldsymbol{x}\sim\mathbb{P}_{S_i}}\left[h(\boldsymbol{x})\neq h'(\boldsymbol{x})\right] - \Pr_{\boldsymbol{x}\sim\mathbb{P}_C}\left[h(\boldsymbol{x})\neq h'(\boldsymbol{x})\right]\right)\right. \\
&\quad \left. + \left(\Pr_{\boldsymbol{x}\sim\mathbb{P}_C}\left[h(\boldsymbol{x})\neq h'(\boldsymbol{x})\right] - \Pr_{\boldsymbol{x}\sim\mathbb{P}_T}\left[h(\boldsymbol{x})\neq h'(\boldsymbol{x})\right]\right)\right| \\
&\leq 2 \sup_{h,h'\in\mathcal{H}} \left|\Pr_{\boldsymbol{x}\sim\mathbb{P}_{S_i}}\left[h(\boldsymbol{x})\neq h'(\boldsymbol{x})\right] - \Pr_{\boldsymbol{x}\sim\mathbb{P}_C}\left[h(\boldsymbol{x})\neq h'(\boldsymbol{x})\right]\right| \\
&\quad + 2 \sup_{h,h'\in\mathcal{H}} \left|\Pr_{\boldsymbol{x}\sim\mathbb{P}_C}\left[h(\boldsymbol{x})\neq h'(\boldsymbol{x})\right] - \Pr_{\boldsymbol{x}\sim\mathbb{P}_T}\left[h(\boldsymbol{x})\neq h'(\boldsymbol{x})\right]\right| \\
&= d_{\mathcal{H}\Delta\mathcal{H}}\left(\mathbb{P}_{S_i}, \mathbb{P}_C\right) + d_{\mathcal{H}\Delta\mathcal{H}}\left(\mathbb{P}_C, \mathbb{P}_T\right)
\end{aligned}
\tag{34}
$$

Since $\mathcal{P}_\alpha = \{\mathbb{P}_\alpha \mid \mathbb{P}_\alpha = \sum_i \alpha_i \mathbb{P}_{S_i},\ \sum_i \alpha_i = 1,\ \alpha_i \geq 0\ \forall i\}$ and $\mathbb{P}_C \in \mathcal{P}_\alpha$, we can obtain that

$$
\begin{aligned}
d_{\mathcal{H}\Delta\mathcal{H}}\left(\mathbb{P}_{S_i}, \mathbb{P}_C\right) &= d_{\mathcal{H}\Delta\mathcal{H}}\left(\mathbb{P}_{S_i}, \sum_j \alpha_j \mathbb{P}_{S_j}\right) \\
&= d_{\mathcal{H}\Delta\mathcal{H}}\left(\sum_j \alpha_j \mathbb{P}_{S_i}, \sum_j \alpha_j \mathbb{P}_{S_j}\right) \\
&\leq \sum_j \alpha_j d_{\mathcal{H}\Delta\mathcal{H}}\left(\mathbb{P}_{S_i}, \mathbb{P}_{S_j}\right) \\
&= d_{\mathcal{H}\Delta\mathcal{H}}\left(\mathbb{P}_{S_i}, \mathbb{P}_{S_j}\right) \\
&\leq \max_{i,j} d_{\mathcal{H}\Delta\mathcal{H}}\left(\mathbb{P}_{S_i}, \mathbb{P}_{S_j}\right)
\end{aligned}
\tag{35}
$$

Combine with Eq. (34) and 35, we have

$$
d_{\mathcal{H}\Delta\mathcal{H}}\left(\mathbb{P}_{S_i}, \mathbb{P}_T\right) \leq \max_{i,j} d_{\mathcal{H}\Delta\mathcal{H}}\left(\mathbb{P}_{S_i}, \mathbb{P}_{S_j}\right) + d_{\mathcal{H}\Delta\mathcal{H}}\left(\mathbb{P}_C, \mathbb{P}_T\right)
\tag{36}
$$

Substitute Eq. (36) into Eq. (33), we obtain that

$$
\begin{aligned}
\epsilon_T(\hat{h}) &\leq \epsilon_T\left(h_T^*\right) + \sum_{i=1}^N \alpha_j\left(2\lambda_i + \max_{i,j} d_{\mathcal{H}\Delta\mathcal{H}}\left(\mathbb{P}_{S_i}, \mathbb{P}_{S_j}\right) + d_{\mathcal{H}\Delta\mathcal{H}}\left(\mathbb{P}_C, \mathbb{P}_T\right)\right) \\
&\quad + 2\sqrt{\left(\sum_{i=1}^N \frac{\alpha_i^2}{\beta_i}\right)\left(\frac{d\log(2m) - \log(\delta)}{2m}\right)}.
\end{aligned}
\tag{37}
$$

Since $\lambda_i = \min_{h\in\mathcal{H}}\left\{\epsilon_T(h) + \epsilon_{S_i}(h)\right\} = \epsilon_T(h^*) + \epsilon_{S_i}(h^*)$, by setting $\lambda = 2\sum_{i=1}^N \alpha_i \epsilon_{S_i}(h^*)$ and $C(\delta, m, d) = 2\sqrt{\left(\sum_{i=1}^N \frac{\alpha_i^2}{\beta_i}\right)\left(\frac{d\log(2m)-\log(\delta)}{2m}\right)}$ yields the proof.

### A.2.3 Discussion

Theorem 2 provides a theoretical framework for estimating performance on a new target distribution. The first term, $\epsilon_T(h_T^*)$, represents the target error under the ideal hypothesis $h_T^*$. The second term, $\lambda = 2\sum_{j=1}^N \alpha_i \epsilon_{S_i}(h)$, aggregates the combined error over all source distributions weighted by $\alpha_i$ and can be minimized via supervised loss with labels. The third term, $d_{\mathcal{H}\Delta\mathcal{H}}\left(\mathbb{P}_C, \mathbb{P}_T\right)$, measures the distributional discrepancy between a composite source distribution and the target distribution. The fourth term, $\max_{i,j} d_{\mathcal{H}\Delta\mathcal{H}}(\mathbb{P}_{S_i}, \mathbb{P}_{S_j})$, quantifies the maximum discrepancy between any two source distributions. The last term, $C(\delta, m, d)$, is a statistical term which depends on the confidence level $\delta$, sample size $m$, and VC dimension $d$.

The tri-level learning framework proposed in Eq. (10) aims to minimize the third term, $d_{\mathcal{H}\Delta\mathcal{H}}\left(\mathbb{P}_C, \mathbb{P}_T\right)$, and the fourth term, $\max_{i,j} d_{\mathcal{H}\Delta\mathcal{H}}(\mathbb{P}_{S_i}, \mathbb{P}_{S_j})$. These two terms correspond to the group-level and sample-level uncertainties, respectively. Below, we discuss how these two terms align with the motivation for our tri-level optimization.

For the term $d_{\mathcal{H}\Delta\mathcal{H}}\left(\mathbb{P}_C, \mathbb{P}_T\right)$, the goal of time series OOD generalization is to learn a model that generalizes well to unseen domain distributions, which makes direct optimization infeasible due to the unavailability of target dataset. To minimize this discrepancy, we can only enlarge the set $\mathcal{P}_C$. Specifically, we manipulate $\delta$-perturbations applied to individual samples in the third level of our tri-level learning framework. By optimizing these perturbations, we explore a broader range

of variations within each source domain, which potentially minimizes the term $d_{\mathcal{H}\Delta\mathcal{H}}\left(\mathbb{P}_C, \mathbb{P}_T\right)$, enhancing the robustness of learned representations.

For the term $\max_{i,j} d_{\mathcal{H}\Delta\mathcal{H}}(\mathbb{P}_{S_i}, \mathbb{P}_{S_j})$, the second-level optimization in our tri-level framework adjusts the weights $\alpha_i$ that define the mixture of source distributions $\mathbb{P}_C$. By dynamically modifying these weights based on the 'worst-case' distribution, we minimize the term $\max_{i,j} d_{\mathcal{H}\Delta\mathcal{H}}(\mathbb{P}_{S_i}, \mathbb{P}_{S_j})$.

Our approach not only enhances representation invariance across diverse domains but also improves the model's resilience against variations within individual samples. The effectiveness of this tri-level framework is rooted in the interdependence between the problems at each level; adjustments in one level influence the conditions and outcomes of the others. This demonstrates the necessity of our tri-level learning optimization, as it requires a coordinated strategy that simultaneously considers sample-level, group-level, and parameter-level dynamics.

### A.3  Proof of Theorem 3

First, the first-order Taylor expansion of $f_2(\boldsymbol{\theta}, \boldsymbol{q}, \boldsymbol{\delta})$ at the point $(\overline{\boldsymbol{\theta}}, \overline{\boldsymbol{\delta}})$ is obtained as follows

$$\tilde{f}_2(\boldsymbol{\theta}, \boldsymbol{q}', \boldsymbol{\delta}) = f_2(\overline{\boldsymbol{\theta}}, \boldsymbol{q}', \overline{\boldsymbol{\delta}}) + \nabla_\theta f_2(\overline{\boldsymbol{\theta}}, \boldsymbol{q}', \overline{\boldsymbol{\delta}})^T(\boldsymbol{\theta} - \overline{\boldsymbol{\theta}}) + \nabla_\delta f_2(\overline{\boldsymbol{\delta}}, \boldsymbol{q}', \overline{\boldsymbol{\delta}})^T(\boldsymbol{\delta} - \overline{\boldsymbol{\delta}}). \tag{38}$$

Since $T_2 = 1$, therefore, we have:

$$\varphi(\boldsymbol{\theta}, \boldsymbol{\delta}) = \boldsymbol{q}'_0 - \eta_q \nabla_{q'} \tilde{f}_2\left(\boldsymbol{\theta}, \boldsymbol{q}', \boldsymbol{\delta}\right). \tag{39}$$

Combine with Eq. (38) and (39), we can obtain that

$$\begin{aligned}
\varphi(\boldsymbol{\theta}, \boldsymbol{\delta}) &= \boldsymbol{q}'_0 - \eta_q \nabla_{q'}\Big(f_2(\overline{\boldsymbol{\theta}}, \boldsymbol{q}', \overline{\boldsymbol{\delta}}) + \nabla_\theta f_2(\overline{\boldsymbol{\theta}}, \boldsymbol{q}', \overline{\boldsymbol{\delta}})(\boldsymbol{\theta} - \overline{\boldsymbol{\theta}}) + \nabla_\delta f_2(\overline{\boldsymbol{\theta}}, \boldsymbol{q}', \overline{\boldsymbol{\delta}})(\boldsymbol{\delta} - \overline{\boldsymbol{\delta}})\Big) \\
&= \boldsymbol{q}'_0 - \eta_q \nabla_{q'} f_2(\overline{\boldsymbol{\theta}}, \boldsymbol{q}', \overline{\boldsymbol{\delta}}) - \eta_q \nabla_{q'}\nabla_\theta f_2(\overline{\boldsymbol{\theta}}, \boldsymbol{q}', \overline{\boldsymbol{\delta}})\boldsymbol{\theta} + \nabla_{q'}\nabla_\theta f_2(\overline{\boldsymbol{\theta}}, \boldsymbol{q}', \overline{\boldsymbol{\delta}})\overline{\boldsymbol{\theta}} \\
&\quad - \nabla_{q'}\nabla_\delta f_2(\overline{\boldsymbol{\theta}}, \boldsymbol{q}', \overline{\boldsymbol{\delta}})\boldsymbol{\delta} + \nabla_{q'}\nabla_\delta f_2(\overline{\boldsymbol{\theta}}, \boldsymbol{q}', \overline{\boldsymbol{\delta}})\overline{\boldsymbol{\delta}} \\
&= -\eta_q \nabla_{q'}\nabla_\theta f_2(\overline{\boldsymbol{\theta}}, \boldsymbol{q}', \overline{\boldsymbol{\delta}})\boldsymbol{\theta} - \nabla_{q'}\nabla_\delta f_2(\overline{\boldsymbol{\theta}}, \boldsymbol{q}', \overline{\boldsymbol{\delta}})\boldsymbol{\delta} + C,
\end{aligned} \tag{40}$$

where $C = \boldsymbol{q}'_0 - \eta_q \nabla_{q'} f_2(\overline{\boldsymbol{\theta}}, \boldsymbol{q}', \overline{\boldsymbol{\delta}}) + \nabla_{q'}\nabla_\theta f_2(\overline{\boldsymbol{\theta}}, \boldsymbol{q}', \overline{\boldsymbol{\delta}})\overline{\boldsymbol{\theta}} + \nabla_{q'}\nabla_\delta f_2(\overline{\boldsymbol{\theta}}, \boldsymbol{q}', \overline{\boldsymbol{\delta}})\overline{\boldsymbol{\delta}}$ is an affine function. Therefore, $\varphi(\boldsymbol{\theta}, \boldsymbol{\delta})$ is a convex function. According to preserve convexity[Boyd and Vandenberghe, 2004], $h(\boldsymbol{\theta}, \boldsymbol{q}, \boldsymbol{\delta}) = \|\boldsymbol{q} - \phi(\boldsymbol{\theta}, \boldsymbol{\delta})\|$ is convexity, which concludes the proof of theorem 3.

### A.4  Proof of Theorem 4

Assume that in the $t^{\text{th}}$ iteration, a new cutting plane is added, and the selected point $(\boldsymbol{\theta}^{t+1}, \boldsymbol{q}^{t+1}, \boldsymbol{\delta}^{t+1})$ always lies within the region $\mathcal{R}_t^c$ formed by the cutting plane set $\mathcal{C}^t$, we have

$$\mathcal{R}_0^c \supseteq \mathcal{R}_1^c \supseteq \cdots \supseteq \mathcal{R}_t^c. \tag{41}$$

Let $\mathcal{H}^t$ denote the feasible region of problem in Eq. (17) at the $t^{\text{th}}$ iteration, and $\mathcal{R}$ represent the feasible region of problem in Eq. (15), then it follows that

$$\mathcal{H}^0 \supseteq \cdots \supseteq \mathcal{H}^t \supseteq \mathcal{H}. \tag{42}$$

Let $F(\boldsymbol{\theta}^t, \boldsymbol{q}^t, \boldsymbol{\delta}^t)$ denote the optimal value of problem in Eq. (17) at the $t^{\text{th}}$ iteration, and $f_1^*$ represent the optimal value of the problem Eq. (15). Based on equation (42), we can obtain

$$F\left(\boldsymbol{\theta}^0, \boldsymbol{q}^0, \boldsymbol{\delta}^0\right) \leq F\left(\boldsymbol{\theta}^t, \boldsymbol{q}^t, \boldsymbol{\delta}^t\right) \leq \cdots \leq F^*. \tag{43}$$

It's seen that the sequence $\left\{F(\boldsymbol{\theta}^t, \boldsymbol{q}^t, \boldsymbol{\delta}^t)\right\}$ is monotonically increasing. As $T_1 \to \infty$, $f_1$ monotonically converges to a certain fixed value. It is worth mentioning that $h(\boldsymbol{\theta}, \boldsymbol{q}, \boldsymbol{\delta}) = \|\boldsymbol{q} - \phi(\boldsymbol{\theta}, \boldsymbol{\delta})\|$ is a convex function. Since the sublevel set of a convex function is convex, the feasible region of problem in Eq. (17) is convexity. This implies that the iterative procedure, by continuously adding a cutting plane, is progressively converging to the optimal value $f_1^*$ of the problem as referenced in Eq. (15).

### A.5  Proof of Theorem 5

To begin with, we introduce a fundamental lemma that is pivotal for the subsequent analysis.

**Lemma 1.** *For* $\frac{1}{m}\sum_{j=1}^m g_\theta\left(\boldsymbol{\theta}^t,\boldsymbol{q}^t,\boldsymbol{\delta}^t;\zeta_j^t\right),\frac{1}{m}\sum_{j=1}^m g_q(\boldsymbol{\theta}^t,\boldsymbol{q}^t,\boldsymbol{\delta}^t;\zeta_j^t),\frac{1}{m}\sum_{j=1}^m g_\delta\left(\boldsymbol{\theta}^t,\boldsymbol{q}^t,\boldsymbol{\delta}^t;\zeta_j^t\right),$ *they are unbiased and bounded, that is,*

$$
\begin{aligned}
\mathbb{E}_{\{\zeta_j^t\}}\left[\tfrac{1}{m}\sum_{j=1}^m g_\theta\left(\boldsymbol{\theta}^t,\boldsymbol{q}^t,\boldsymbol{\delta}^t;\zeta_j^t\right)\right] &= \nabla_\theta f_1\left(\boldsymbol{\theta}^t,\boldsymbol{q}^t,\boldsymbol{\delta}^t\right),\\
\mathbb{E}_{\{\zeta_j^t\}}\left[\left\|\tfrac{1}{m}\sum_{j=1}^m g_\theta\left(\boldsymbol{\theta}^t,\boldsymbol{q}^t,\boldsymbol{\delta}^t;\zeta_j^t\right)\right\|^2\right] &\le \alpha_1^2 + \tfrac{\sigma_1^2}{m},
\end{aligned}
\tag{44}
$$

$$
\begin{aligned}
\mathbb{E}_{\{\zeta_j^t\}}\left[\tfrac{1}{m}\sum_{j=1}^m g_q\left(\boldsymbol{\theta}^t,\boldsymbol{q}^t,\boldsymbol{\delta}^t;\zeta_j^t\right)\right] &= \nabla_q f_1\left(\boldsymbol{\theta}^t,\boldsymbol{q}^t,\boldsymbol{\delta}^t\right),\\
\mathbb{E}_{\{\zeta_j^t\}}\left[\left\|\tfrac{1}{m}\sum_{j=1}^m g_q\left(\boldsymbol{\theta}^t,\boldsymbol{q}^t,\boldsymbol{\delta}^t;\zeta_j^t\right)\right\|^2\right] &\le \alpha_2^2 + \tfrac{\sigma_2^2}{m},
\end{aligned}
\tag{45}
$$

$$
\begin{aligned}
\mathbb{E}_{\{\zeta_j^t\}}\left[\tfrac{1}{m}\sum_{j=1}^m g_\delta\left(\boldsymbol{\theta}^t,\boldsymbol{q}^t,\boldsymbol{\delta}^t;\zeta_j^t\right)\right] &= \nabla_\delta f_1\left(\boldsymbol{\theta}^t,\boldsymbol{q}^t,\boldsymbol{\delta}^t\right),\\
\mathbb{E}_{\{\zeta_j^t\}}\left[\left\|\tfrac{1}{m}\sum_{j=1}^m g_\delta\left(\boldsymbol{\theta}^t,\boldsymbol{q}^t,\boldsymbol{\delta}^t;\zeta_j^t\right)\right\|^2\right] &\le \alpha_3^2 + \tfrac{\sigma_3^2}{m}.
\end{aligned}
\tag{46}
$$

Here, $\mathbb{E}_{\{\zeta_j^t\}}[\cdot]$ denotes the expectation with respect to a set of variables $\{\zeta_1^t,\cdots,\zeta_m^t\}$.

### A.5.1 Proof of Lemma 1

Taking the variable $\boldsymbol{\theta}$ as an example, according to Assumption 4, for all $i = 1,\cdots,n$, we have

$$
\mathbb{E}_{\{\zeta_j^t\}}\left[\tfrac{1}{m}\sum_{j=1}^m g_\theta\left(\boldsymbol{\theta}^t,\boldsymbol{q}^t,\boldsymbol{\delta}^t;\zeta_j^t\right)\right] = \tfrac{1}{m}\sum_{j=1}^m \mathbb{E}_{\zeta_j^t}\left[g_\theta\left(\boldsymbol{\theta}^t,\boldsymbol{q}^t,\boldsymbol{\delta}^t;\zeta_j^t\right)\right] = \nabla_\theta F\left(\boldsymbol{\theta}^t,\boldsymbol{q}^t,\boldsymbol{\delta}^t\right).
\tag{47}
$$

Based on Assumption 5, the variance of $\nabla_\theta F(\boldsymbol{\theta}^t,\boldsymbol{q}^t,\boldsymbol{\delta}^t)$ is bounded, from which we can deduce

$$
\begin{aligned}
&\mathbb{E}_{\{\zeta_j^t\}}\left[\tfrac{1}{m}\sum_{j=1}^m g_\theta\left(\boldsymbol{\theta}^t,\boldsymbol{q}^t,\boldsymbol{\delta}^t;\zeta_j^t\right)\right]\\
&= \mathbb{E}_{\{\zeta_j\}}\left[\left\|\tfrac{1}{m}\sum_{j=1}^m g_\theta\left(\boldsymbol{\theta}^t,\boldsymbol{q}^t,\boldsymbol{\delta}^t;\zeta_j^t\right) - \nabla_\theta F\left(\boldsymbol{\theta}^t,\boldsymbol{q}^t,\boldsymbol{\delta}^t\right)\right\|^2\right] + \left\|\nabla_\theta F\left(\boldsymbol{\theta}^t,\boldsymbol{q}^t,\boldsymbol{\delta}^t\right)\right\|^2\\
&= \tfrac{\sum_{j=1}^m \mathbb{E}_{\zeta_j}\left[\|g_\theta(\boldsymbol{\theta},\boldsymbol{q},\boldsymbol{\delta};\zeta_j) - \nabla_\theta F(\boldsymbol{\theta},\boldsymbol{q},\boldsymbol{\delta})\|^2\right]}{m^2} + \alpha_1^2\\
&\le \tfrac{\sigma_1^2}{m} + \alpha_1^2.
\end{aligned}
\tag{48}
$$

The proofs of Eq. (45) and Eq. (46) follow a similar logic. Thus, we complete the proof of Lemma 1.

Combine with lemma 1, we now proceed to derive Theorem 5. Under Assumption 3 that the gradient of $F$ is Lipschitz continuous, for $t > t_1$, it follows that

$$
\begin{aligned}
&F\left(\boldsymbol{\theta}^{t+1},\boldsymbol{q}^{t+1},\boldsymbol{\delta}^{t+1}\right)\\
&\le F\left(\boldsymbol{\theta}^t,\boldsymbol{q}^t,\boldsymbol{\delta}^t\right) + \left\langle\nabla_\theta F\left(\boldsymbol{\theta}^t,\boldsymbol{q}^t,\boldsymbol{\delta}^t\right),\boldsymbol{\theta}^{t+1}-\boldsymbol{\theta}^t\right\rangle + \left\langle\nabla_q F\left(\boldsymbol{\theta}^t,\boldsymbol{q}^t,\boldsymbol{\delta}^t\right),\boldsymbol{q}^{t+1}-\boldsymbol{q}^t\right\rangle\\
&\quad + \left\langle\nabla_\delta F\left(\boldsymbol{\theta}^t,\boldsymbol{q}^t,\boldsymbol{\delta}^t\right),\boldsymbol{\delta}^{t+1}-\boldsymbol{\delta}^t\right\rangle + \tfrac{L}{2}\left(\left\|\boldsymbol{\theta}^{t+1}-\boldsymbol{\theta}^t\right\|^2 + \left\|\boldsymbol{q}^{t+1}-\boldsymbol{q}^t\right\|^2 + \left\|\boldsymbol{\delta}^{t+1}-\boldsymbol{\delta}^t\right\|^2\right)\\
&= F\left(\boldsymbol{\theta}^t,\boldsymbol{q}^t,\boldsymbol{\delta}^t\right) - \eta_\theta\left\langle\nabla_\theta F\left(\boldsymbol{\theta}^t,\boldsymbol{q}^t,\boldsymbol{\delta}^t\right),\tfrac{1}{m}\sum_{j=1}^m g_\theta\left(\boldsymbol{\theta}^t,\boldsymbol{q}^t,\boldsymbol{\delta}^t;\zeta_j^t\right)\right\rangle\\
&\quad -\eta_q\left\langle\nabla_q F\left(\boldsymbol{\theta}^t,\boldsymbol{q}^t,\boldsymbol{\delta}^t\right),\tfrac{1}{m}\sum_{j=1}^m g_q\left(\boldsymbol{\theta}^t,\boldsymbol{q}^t,\boldsymbol{\delta}^t;\zeta_j^t\right)\right\rangle\\
&\quad -\eta_\delta\left\langle\nabla_\delta F\left(\boldsymbol{\theta}^t,\boldsymbol{q}^t,\boldsymbol{\delta}^t\right),\tfrac{1}{m}\sum_{j=1}^m g_\delta\left(\boldsymbol{\theta}^t,\boldsymbol{q}^t,\boldsymbol{\delta}^t;\zeta_j^t\right)\right\rangle\\
&\quad +\tfrac{L(\eta_\theta)^2}{2}\left\|\tfrac{1}{m}\sum_{j=1}^m g_\theta\left(\boldsymbol{\theta}^t,\boldsymbol{q}^t,\boldsymbol{\delta}^t;\zeta_j^t\right)\right\|^2\\
&\quad +\tfrac{L(\eta_q)^2}{2}\left\|\tfrac{1}{m}\sum_{j=1}^m g_q\left(\boldsymbol{\theta}^t,\boldsymbol{q}^t,\boldsymbol{\delta}^t;\zeta_j^t\right)\right\|^2\\
&\quad +\tfrac{L(\eta_\delta)^2}{2}\left\|\tfrac{1}{m}\sum_{j=1}^m g_\delta\left(\boldsymbol{\theta}^t,\boldsymbol{q}^t,\boldsymbol{\delta}^t;\zeta_j^t\right)\right\|^2.
\end{aligned}
\tag{49}
$$

Taking the expectation of both sides of Equation (49) with respect to $\{\zeta_1^t, \cdots, \zeta_m^t\}$, we can obtain

$$
\mathbb{E}_{\{\zeta_j^t\}}\left[F\left(\boldsymbol{\theta}^{t+1}, \boldsymbol{q}^{t+1}, \boldsymbol{\delta}^{t+1}\right)\right]
$$

$$
\leq F\left(\boldsymbol{\theta}^t, \boldsymbol{q}^t, \boldsymbol{\delta}^t\right) - \eta_\theta \mathbb{E}_{\{\zeta_j^t\}}\left\langle \nabla_\theta F\left(\boldsymbol{\theta}^t, \boldsymbol{q}^t, \boldsymbol{\delta}^t\right), \frac{1}{m}\sum\nolimits_{j=1}^m g_\theta\left(\boldsymbol{\theta}^t, \boldsymbol{q}^t, \boldsymbol{\delta}^t; \zeta_j^t\right)\right\rangle
$$

$$
- \eta_q \mathbb{E}_{\{\zeta_j^t\}}\left\langle \nabla_q F\left(\boldsymbol{\theta}^t, \boldsymbol{q}^t, \boldsymbol{\delta}^t\right), \frac{1}{m}\sum\nolimits_{j=1}^m g_q\left(\boldsymbol{\theta}^t, \boldsymbol{q}^t, \boldsymbol{\delta}^t; \zeta_j^t\right)\right\rangle
$$

$$
- \eta_\delta \mathbb{E}_{\{\zeta_j^t\}}\left\langle \nabla_\delta F\left(\boldsymbol{\theta}^t, \boldsymbol{q}^t, \boldsymbol{\delta}^t\right), \frac{1}{m}\sum\nolimits_{j=1}^m g_\delta\left(\boldsymbol{\theta}^t, \boldsymbol{q}^t, \boldsymbol{\delta}^t; \zeta_j^t\right)\right\rangle
$$

$$
+ \frac{L\left(\eta_\theta\right)^2}{2}\mathbb{E}_{\{\zeta_j^t\}}\left\|\frac{1}{m}\sum\nolimits_{j=1}^m g_\theta\left(\boldsymbol{\theta}^t, \boldsymbol{q}^t, \boldsymbol{\delta}^t; \zeta_j^t\right)\right\|^2
$$

$$
+ \frac{L\left(\eta_q\right)^2}{2}\mathbb{E}_{\{\zeta_j^t\}}\left\|\frac{1}{m}\sum\nolimits_{j=1}^m g_q\left(\boldsymbol{\theta}^t, \boldsymbol{q}^t, \boldsymbol{\delta}^t; \zeta_j^t\right)\right\|^2
$$

$$
+ \frac{L\left(\eta_\delta\right)^2}{2}\mathbb{E}_{\{\zeta_{\tilde{j}}^t\}}\left\|\frac{1}{m}\sum\nolimits_{j=1}^m g_\delta\left(\boldsymbol{\theta}^t, \boldsymbol{q}^t, \boldsymbol{\delta}^t; \zeta_j^t\right)\right\|^2
$$

$$
\overset{(i)}{\leq} F\left(\boldsymbol{\theta}^t, \boldsymbol{q}^t, \boldsymbol{\delta}^t\right) - \left\|\nabla_\theta F\left(\boldsymbol{\theta}^t, \boldsymbol{q}^t, \boldsymbol{\delta}^t\right)\right\|^2 \eta_\theta - \left\|\nabla_q F\left(\boldsymbol{\theta}^t, \boldsymbol{q}^t, \boldsymbol{\delta}^t\right)\right\|^2 \eta_q - \left\|\nabla_\delta F\left(\boldsymbol{\theta}^t, \boldsymbol{q}^t, \boldsymbol{\delta}^t\right)\right\|^2 \eta_\delta
$$

$$
+ \frac{L\left(\eta_\theta\right)^2}{2}\left(\alpha_1^2 + \frac{\sigma_1^2}{m}\right) + \frac{L\left(\eta_q\right)^2}{2}\left(\alpha_2^2 + \frac{\sigma_2^2}{m}\right) + \frac{L\left(\eta_\delta\right)^2}{2}\left(\alpha_3^2 + \frac{\sigma_3^2}{m}\right).
$$

(50)

The inequality (i) is based on lemma 1. Taking the total expectation of both sides of Eq. (50), we have

$$
\mathbb{E}\left[F\left(\boldsymbol{\theta}^{t+1}, \boldsymbol{q}^{t+1}, \boldsymbol{\delta}^{t+1}\right)\right]
$$

$$
\leq \mathbb{E}\left[F\left(\boldsymbol{\theta}^t, \boldsymbol{q}^t, \boldsymbol{\delta}^t\right)\right] - \eta_\theta \mathbb{E}\left[\left\|\nabla_\theta F\left(\boldsymbol{\theta}^t, \boldsymbol{q}^t, \boldsymbol{\delta}^t\right)\right\|^2\right] - \eta_q \mathbb{E}\left[\left\|\nabla_q F\left(\boldsymbol{\theta}^t, \boldsymbol{q}^t, \boldsymbol{\delta}^t\right)\right\|^2\right]
$$

$$
- \eta_\delta \mathbb{E}\left[\left\|\nabla_\delta F\left(\boldsymbol{\theta}^t, \boldsymbol{q}^t, \boldsymbol{\delta}^t\right)\right\|^2\right] + \frac{L\left(\eta_\theta\right)^2}{2}\left(\alpha_1^2 + \frac{\sigma^2}{m}\right) + \frac{L\left(\eta_q\right)^2}{2}\left(\alpha_2^2 + \frac{\sigma^2}{m}\right)
$$

$$
+ \frac{L\left(\eta_\delta\right)^2}{2}\left(\alpha_3^2 + \frac{\sigma^2}{m}\right),
$$

(51)

where $\mathbb{E}[\cdot]$ denotes the expectation over all terms. Summing Eq. (51) from $t = t_1$ to $t = T_1 - 1$, we obtain

$$
\mathbb{E}\left[F\left(\boldsymbol{\theta}^{T_1}, \boldsymbol{q}^{T_1}, \boldsymbol{\delta}^{T_1}\right)\right]
$$

$$
\leq \mathbb{E}\left[F\left(\boldsymbol{\theta}^{t_1}, \boldsymbol{q}^{t_1}, \boldsymbol{\delta}^{t_1}\right)\right] - \eta_\theta \sum_{t=t_1}^{T_1-1} \mathbb{E}\left[\left\|\nabla_\theta F\left(\boldsymbol{\theta}^t, \boldsymbol{q}^t, \boldsymbol{\delta}^t\right)\right\|^2\right] - \eta_q \sum_{t=T_1}^{T_1-1} \mathbb{E}\left[\left\|\nabla_q F\left(\boldsymbol{\theta}^t, \boldsymbol{q}^t, \boldsymbol{\delta}^t\right)\right\|^2\right]
$$

$$
- \eta_\delta \sum_{t=t_1}^{T_1-1} \mathbb{E}\left[\left\|\nabla_\delta F\left(\boldsymbol{\theta}^t, \boldsymbol{q}^t, \boldsymbol{\delta}^t\right)\right\|^2\right] + \sum_{t=t_1}^{T_1-1} \frac{L\left(\eta_\theta\right)^2}{2}\left(\alpha_1^2 + \frac{\sigma_1^2}{m}\right)
$$

$$
+ \sum_{t=t_1}^{T_1-1} \frac{L\left(\eta_q\right)^2}{2}\left(\alpha_2^2 + \frac{\sigma_2^2}{m}\right) + \sum_{t=t_1}^{T_1-1} \frac{L\left(\eta_\delta\right)^2}{2}\left(\alpha_3^2 + \frac{\sigma_3^2}{m}\right)
$$

$$
= \mathbb{E}\left[F\left(\boldsymbol{\theta}^{t_1}, \boldsymbol{q}^{t_1}, \boldsymbol{\delta}^{t_1}\right)\right] - \eta_\theta \sum_{t=t_1}^{T_1-1} \mathbb{E}\left[\left\|\nabla_\theta F\left(\boldsymbol{\theta}^t, \boldsymbol{q}^t, \boldsymbol{\delta}^t\right)\right\|^2\right] - \eta_q \sum_{t=T_1}^{T_1-1} \mathbb{E}\left[\left\|\nabla_q F\left(\boldsymbol{\theta}^t, \boldsymbol{q}^t, \boldsymbol{\delta}^t\right)\right\|^2\right]
$$

$$
- \eta_\delta \sum_{t=t_1}^{T_1-1} \mathbb{E}\left[\left\|\nabla_\delta F\left(\boldsymbol{\theta}^t, \boldsymbol{q}^t, \boldsymbol{\delta}^t\right)\right\|^2\right] + \frac{L}{2} \sum_{t=t_1}^{T_1-1} (\eta_\theta^2 \alpha_1^2 + \eta_q^2 \alpha_2^2 + \eta_\delta^2 \alpha_3^2)
$$

$$
+ \frac{L}{2m} \sum_{t=t_1}^{T_1-1} (\eta_\theta^2 \sigma_1^2 + \eta_q^2 \sigma_2^2 + \eta_\delta^2 \sigma_3^2)
$$

(52)

Let $\eta_\theta = \eta_q = \eta_\delta = \frac{1}{\sqrt{T_1 - t_1}}$, Combining with Eq. (52), we have that

$$
\sum_{t=t_1}^{T_1-1} \mathbb{E}\left[\left\|\nabla_\theta F\left(\boldsymbol{\theta}^t, \boldsymbol{q}^t, \boldsymbol{\delta}^t\right)\right\|^2 + \left\|\nabla_q F\left(\boldsymbol{\theta}^t, \boldsymbol{q}^t, \boldsymbol{\delta}^t\right)\right\|^2 + \left\|\nabla_\delta F\left(\boldsymbol{\theta}^t, \boldsymbol{q}^t, \boldsymbol{\delta}^t\right)\right\|^2\right]
$$

$$
\leq F\left(\boldsymbol{\theta}^{T_1}, \boldsymbol{q}^{T_1}, \boldsymbol{\delta}^{T_1}\right) - F^* + \frac{L}{2}(T_1 - t_1)^{-\frac{1}{2}} \sum_{t=t_1}^{T_1-1} \sum_{i=1}^3 \alpha_i^2
$$

$$
+ \frac{L}{2m}(T_1 - t_1)^{-\frac{1}{2}} \sum_{t=t_1}^{T_1-1} \sum_{i=1}^3 \sigma_i^2.
$$

(53)

Combining the definition of $\epsilon$-stationary point described in Definition (3) with Eq. (53), we have

$$
\sum_{t=t_1}^{T_1-1} \mathbb{E}\left[\left\|\nabla_\theta F\left(\boldsymbol{\theta}^t, \boldsymbol{q}^t, \boldsymbol{\delta}^t\right)\right\|^2 + \left\|\nabla_q F\left(\boldsymbol{\theta}^t, \boldsymbol{q}^t, \boldsymbol{\delta}^t\right)\right\|^2 + \left\|\nabla_\delta F\left(\boldsymbol{\theta}^t, \boldsymbol{q}^t, \boldsymbol{\delta}^t\right)\right\|^2\right]
$$

$$
\leq F\left(\boldsymbol{\theta}^{T_1}, \boldsymbol{q}^{T_1}, \boldsymbol{\delta}^{T_1}\right) - f_1^* + \frac{L}{2}(T_1(\epsilon) - t_1)^{-\frac{1}{2}} \sum_{t=t_1}^{T_1-1} \sum_{i=1}^{3} \alpha_i^2
$$
$$
+ \frac{L}{2m}(T_1(\epsilon) - t_1)^{-\frac{1}{2}} \sum_{t=t_1}^{T_1-1} \sum_{i=1}^{3} \sigma_i^2 \tag{54}
$$
$$
\leq \epsilon,
$$

that is,

$$
T_1(\epsilon) \sim t_1 + \frac{L^2(m(\alpha_1^2 + \alpha_2^2 + \alpha_3^2) + \sigma_1^2 + \sigma_2^2 + \sigma_3^2)^2}{4m^2(\epsilon - F(\boldsymbol{\theta}^{T_1}, \boldsymbol{q}^{T_1}, \boldsymbol{\delta}^{T_1}) + F^*)^2}. \tag{55}
$$

According to Eq. (55), we can obtain

$$
T_1(\epsilon) \sim \frac{1}{\epsilon^2} \tag{56}
$$

Hence, it can be concluded that there exists a $T_1(\epsilon)$ such that $\|\nabla G^t\| = \mathbb{E}\left[\left\|\nabla_\theta F\left(\boldsymbol{\theta}^t, \boldsymbol{q}^t, \boldsymbol{\delta}^t\right)\right\|^2 + \left\|\nabla_q F\left(\boldsymbol{\theta}^t, \boldsymbol{q}^t, \boldsymbol{\delta}^t\right)\right\|^2 + \left\|\nabla_\delta F\left(\boldsymbol{\theta}^t, \boldsymbol{q}^t, \boldsymbol{\delta}^t\right)\right\|^2\right] \leq \epsilon$, which concludes the proof of Theorem 5.

# B    Datasets

## B.1    Datasets Information

**HHAR**[Blunck et al., 2015] collected activity data from 9 subjects engaging in 6 activities, using smartphones that capture 3D accelerometer data from various positions. **PAMAP**[Reiss, 2012] encompasses 18 physical activities data from 9 subjects, recorded using wearable sensors monitoring physiological and movement metrics. **WESAD**[Philip Schmidt et al., 2018] focuses on stress and affective state detection from 15 subjects, employing wearable sensors for ECG, EMG, respiration, and temperature under varied conditions. **SWELL**[Koldijk et al., 2014] recorded stress responses in a work environment from 25 participants using ECG, EDA, and heart rate sensors during typical office tasks under stressors. **USC-HAD**[Zhang and Sawchuk, 2012] comprises detailed motion and orientation data from 14 subjects performing various activities, captured via a MotionNode device with high sampling rate. **DSADS**[Barshan and Altun, 2013] consists of 19 activities recorded from 8 subjects at Bilkent University, using body-worn sensors on torso and limbs, with data segmented for detailed analysis. Table 2 shows the information of the 6 datasets we used in our experiments.

Table 2: Dataset information.

| Dataset | Classes | Dimension | subjects | samples |
|---|---|---|---|---|
| HHAR | 6 | 3 | 4 | 73420 |
| PAMAP | 13 | 40 | 9 | 31984 |
| WESAD | 4 | 8 | 4 | 63180 |
| SWELL | 2 | 3 | 16 | 52944 |
| USC-HAD | 12 | 3 | 4 | 40244 |
| DSADS | 19 | 45 | 4 | 17520 |

## B.2    Domain Setting

The domain setting is summarized in Table 3. This setting is done to balance the number of samples and classed across different domains.

## B.3    Data pre-processing

For all datasets, we configure the window size as 128 and the step size as 64, resulting in a 50% overlap between two adjacent time series samples. Each sample is standardized using the formula $\tilde{x} = \frac{x-\mu}{\sigma}$, where $\mu$ and $\sigma$ represent the mean and standard deviation of the dataset, respectively. It's

Table 3: Domain setting for HHAR, PAMAP and WESAD dataset.

| Dataset | Domain A | Domain B | Domain C | Domain D |
|---------|----------|----------|----------|----------|
| HHAR | 1 | 2 | 3 | 4 |
| PAMAP | 1 | 2 | 3 | 4 |
| WESAD | 1,2 | 3,5 | 4,6,9 | 7,8 |
| SWELL | 1,2,4,5 | 6,7,9,10 | 12,13,14,16 | 17,18,21,24 |
| USC-HAD | 2,3,4 | 4,5,6 | 1,7,9,10 | 11,12,13,14 |
| DSADS | 1,2 | 3,4 | 5,6 | 7,8 |

important to note that the $\mu$ and $\sigma$ used here is specific to each domain, rather than the entire dataset. This approach is intended to maximize the distinction between data from different domains.

## C   Network Architecture and Hyperparameters

Our baseline experiments were conducted using a network architecture consisting of 10-layers dilated convolutions network. The dilation rate for each layer is set to $2^k$, where $k$ is the layer number. We used the same kernel size of 3 across all layers. Optimization was performed using the Adam optimizer with a weight decay of $3 \times 10^{-4}$. For all baseline experiments, we set the batch size to 256 and the learning rate to 0.002. The training was set to run for a maximum of 50 epochs. All the methods are implemented with PyTorch[Paszke et al., 2019] version 1.7.1 on an NVIDIA GeForce RTX 4090 graphics card.

In the TTSO fine-tuning experiments, we employed GPT-2 as the language model. Fine-tuning was performed in two stage: In the first stage, the learning rate for the large model was $1 \times 10^{-4}$, and for the input embedding, it was set at 0.001. In the second phase, we adjusted the learning rate for the large model to $5 \times 10^{-5}$, aiming to further refine the model's performance. During the evaluation phase, we froze the parameters of the language model, only fine-tuning the classifier with a learning rate of 0.003 to adapt to the specific classification tasks. The batch size was consistently set at 16 for all experimental stages.

## D   Details of Fine-tuning

The structure of LLM Fine-tuning with TTSO is illustrated in figure 3a and 3b. The primary components involved are as follows:

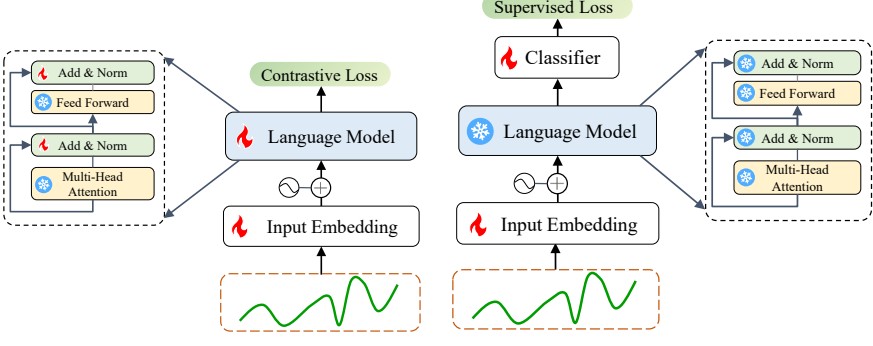

(a) First Stage: Alignment Fine-tuning          (b) Second Stage: Downstream Fine-tuning

Figure 3: Structure of LLM Fine-tuning with TTSO, illustrating the two-phase approach starting with alignment fine-tuning followed by downstream fine-tuning, adapted specifically for time series out-of-distribution generalization tasks.

**Input Embedding**: The first is the input embedding, where raw time series is transformed into embedding space that is amenable for processing by the language model. In our experiments, the input embedding is a linear layer. As shown in Figure 3, the time series (indicated by the dashed box below the waveform) is combined with positional encoding to form the input representation.

**Language Model**: This is the core part of the model, typically comprising multiple pretrained transformer encoder. This model is used for processing the input embeddings and producing advanced

feature representations for subsequent classification tasks. Note that, to retain the intrinsic information of the language model, only the parameters of layer normalization can be tuned.

**Classifier**: In the second stage of fine-tuning, the classifier tailors the language model to the specific time series classification task. It takes the advanced feature representations from the language model and fine-tunes the model for downstream tasks. We use a linear layer for the classifier, with cross entropy as the supervised loss.

**Contrastive Loss**: A contrastive loss function is employed to enhance the discriminative of the representations in the first stage of the fine-tuning process. This loss function aims to ensure that the representations of similar time series samples are brought closer together in the representations space, while representations of dissimilar samples are pushed apart. Specifically, during this stage, the contrastive loss acts as a guiding signal for the language model, encouraging it to learn representations that effectively capture the underlying patterns and distinctions within the time series data, thereby adapting the language model to time series data more effectively.

# E   More Experiments

## E.1   Additional Datasets

We conducte on more datasets to demonstrate the superior performance of our framework. The results is summarized in Table 4.

Table 4: Classification accuracy(%) on SWELL, USC-HAD and DSADS datasets.

| Method | SWELL | | | | | USC-HAD | | | | | DSADS | | | | | ALL |
|---|---|---|---|---|---|---|---|---|---|---|---|---|---|---|---|---|
| | A | B | C | D | AVG | A | B | C | D | AVG | A | B | C | D | AVG | AVG |
| ADARNN | 58.3 | 66.2 | 57.5 | 50.6 | 58.2 | 59.5 | 60.7 | 60.0 | 58.6 | 59.7 | 88.1 | 77.5 | 91.3 | 82.9 | 85.0 | 67.7 |
| GILE | 56.7 | 58.1 | 54.9 | 62.3 | 58.0 | 61.0 | 64.3 | 66.2 | 57.2 | 62.2 | 84.7 | 76.3 | 82.6 | 78.2 | 80.5 | 66.9 |
| ERM | 55.6 | 59.6 | 53.0 | 61.1 | 57.3 | 60.7 | 62.9 | 64.1 | 59.3 | 61.8 | 86.7 | 81.9 | 87.3 | 81.7 | 84.4 | 67.9 |
| IRM | 60.4 | 58.5 | 52.4 | 60.3 | 57.9 | 60.3 | 52.3 | 68.2 | 56.3 | 59.3 | 89.9 | 78.4 | 90.1 | 83.1 | 85.4 | 67.5 |
| GroupDRO | 61.9 | 60.8 | 51.5 | 59.1 | 58.3 | 62.6 | 63.3 | 67.6 | 58.3 | 63.0 | 92.0 | 81.6 | 90.0 | 82.6 | 86.6 | 69.3 |
| ANDMask | 58.1 | 57.9 | 52.0 | 63.0 | 57.8 | 57.8 | 58.4 | 66.3 | 57.6 | 60.0 | 89.7 | 79.0 | 89.9 | 82.5 | 85.3 | 67.7 |
| RSC | 55.6 | 60.5 | 65.3 | 58.1 | 59.9 | 56.7 | 56.3 | 66.4 | 56.6 | 59.0 | 85.0 | 81.9 | 88.1 | 80.9 | 84.0 | 67.6 |
| Mixup | 57.5 | 61.5 | 57.2 | 52.6 | 57.2 | 64.9 | 61.4 | 67.9 | 56.4 | 62.9 | 93.9 | 82.3 | 91.5 | 84.7 | 88.1 | 69.4 |
| VERx | 57.3 | 58.2 | 51.7 | 57.5 | 56.2 | 58.6 | 57.3 | 66.5 | 59.1 | 60.4 | 89.8 | 79.0 | 95.4 | 87.0 | 87.9 | 68.1 |
| DIFEX | 55.7 | 61.0 | 61.6 | 57.2 | 58.9 | 59.8 | 56.5 | 67.2 | 56.7 | 60.1 | 87.8 | 82.8 | 91.0 | 83.2 | 86.2 | 68.4 |
| DIVERSIFY | 62.8 | 61.9 | 59.2 | 63.4 | 61.8 | 64.7 | 62.2 | 60.3 | 65.1 | 63.1 | 89.0 | 86.2 | 92.2 | 85.7 | 88.3 | 71.3 |
| TTSO | 65.2 | 65.7 | 66.5 | 59.1 | 64.6 | 63.1 | 61.2 | 68.0 | 66.0 | 64.6 | 93.0 | 88.9 | 91.7 | 87.8 | 90.4 | 73.2 |
| TTSO* | 60.5 | 68.7 | 72.4 | 63.5 | 66.3 | 62.1 | 63.5 | 65.2 | 63.0 | 63.3 | 92.0 | 89.9 | 92.2 | 89.5 | 90.9 | 73.4 |

## E.2   Illustration of Sample-level and Group-level Uncertainties

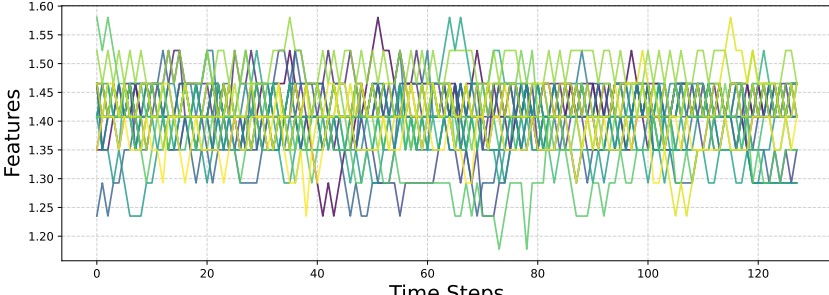

Figure 4: Sample-Level Uncertainty: Each line represents a window of time series data with the same label.

To illustrate the concepts of sample-level and group-level uncertainties, we use the x-axis values from accelerometer data collected by the 'samsungold_1' device from four users in the HHAR dataset.

In Figure 4, sample-level uncertainty is shown by plotting time series data from a specific label (e.g., 'walking'), where each line represents a different time window. The variations among these lines illustrate the inherent noise, which represents sample-level uncertainty.

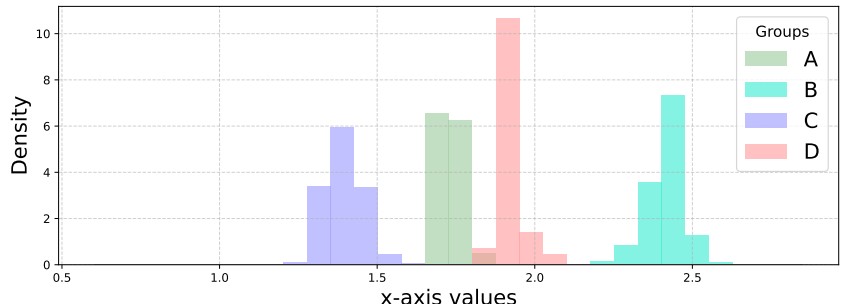

Figure 5: Group-Level Uncertainty: Histogram of 'x' axis values, with each color representing a different group.

Figure 5 demonstrates the group-level uncertainty by displaying the distribution of x-axis values from the accelerometer across different groups (users). Each color represents a distinct group, and each group's unique characteristics contribute to the overall group-level uncertainty.

### E.3 Ablation Study on LLM Architectures and Parameter Configurations

To further investigate the impact of various architectures and parameter settings of LLMs, we conducted additional ablation experiments that focused on different LLM architectures and parameter sizes (e.g., base model and large model). These experiments included encoder-only models (e.g., BERT), decoder-only models (e.g., GPT-2), and encoder-decoder models (e.g., BART) to determine which configurations yield the greatest benefits during fine-tuning. The results are summarized in the table 5.

Table 5: Performance comparison of different LLM architectures and parameter sizes across datasets.

| Architecture | Version | HHAR | PAMAP | WESAD | AVG |
|---|---|---|---|---|---|
| **Encoder-Only (BERT)** | Base | 64.3 | 66.9 | 64.4 | 64.2 |
| | Large | 61.7 | 52.5 | 62.3 | 58.8 |
| **Decoder-Only (GPT)** | Base | 72.9 | 76.1 | 68.4 | 72.5 |
| | Large | 64.5 | 69.4 | 66.5 | 66.8 |
| **Encoder-Decoder (BART)** | Base | 57.3 | 65.4 | 64.2 | 62.3 |
| | Large | 55.5 | 61.2 | 61.4 | 59.4 |

The results indicate that decoder-only architectures, specifically the GPT-2 base model in this experiments, achieve the best performance. However, increasing the number of parameters in all three architectures leads to a significant drop in performance across 3 architectures for time series OOD generalization.

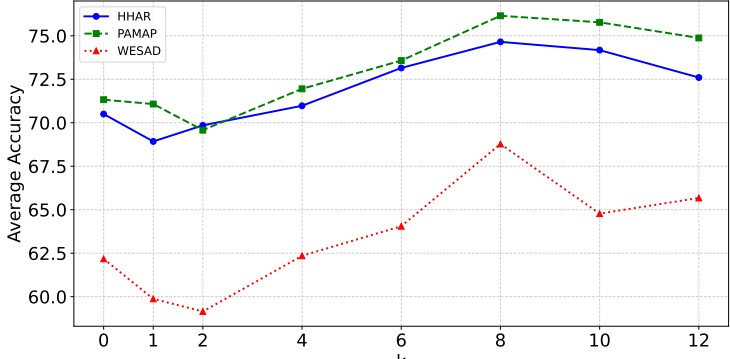

Figure 6: The effect of varying the number of Transformer layers (k) on average accuracy for OOD generalization across HHAR, PAMAP, and WESAD datasets.

To further explore how the number of parameters affects performance, we conducted experiments using GPT-2 models with varying numbers of Transformer layers on 3 datasets to evaluate their OOD

generalization performance. For these experiments, we utilized 20% of each dataset. As shown in Figure 3 (in the attached PDF), the results demonstrate that optimal OOD generalization performance is achieved with a configuration of 8 Transformer layers.

Based on this findings, we incorporate this optimal layer configuration with TTSO framework, yielding improved results as detailed in the table 6.

Table 6: Performance improvements on different domains for HHAR, PAMAP, and WESAD datasets.

| Target | A | B | C | D | AVG |
|--------|------|------|------|------|-------|
| HHAR | +1.4 | +0.1 | +0.8 | +0.9 | +0.80 |
| PAMAP | +1.0 | -1.4 | +0.8 | -0.3 | +0.03 |
| WESAD | +5.5 | -3.5 | +2.5 | -0.1 | +1.35 |

# F   Limitation

TTSO is a general framework for learning invariant representations across diverse domain distributions, currently discussed only for time series classification. This framework could be further enhanced by extending it to more time series OOD tasks, such as time series forecasting and anomaly detection. Additionally, distribution shifts occur not only in time series but also in other machine learning domains like images [Deecke et al., 2021] and text [Tan et al., 2022]. Applying our approach to these domains could further improve performance.

