# OpenReview forum: "Tri-Level Navigator: LLM-Empowered Tri-Level Learning for Time Series OOD Generalization"
_NeurIPS.cc/2024/Conference — NeurIPS 2024 poster_

### Official Review · Reviewer_9773 · 2024-07-11

**Soundness:** 3
**Presentation:** 2
**Contribution:** 3
**Rating:** 6
**Confidence:** 2

**Summary:**

The paper presents a novel approach for time series out-of-distribution generalization via pre-trained large language models.
The authors introduce a Tri-level learning framework that combines sample-level and group-level uncertainties accompanied by a theoretic perspective. Furthermore, a stratified localization algorithm is proposed for the tri-level optimization problem followed by a theoretical convergence guarantee. The paper demonstrates performance gain in six real-world time series datasets, demonstrating the effectiveness of the method.

**Strengths:**

1. The method proposed by the authors is sensible and seems to perform well compared to other out-of-distribution generalization methods.

2. Authors provide an in-depth analysis of their method and provide a convergence analysis.

3. The paper conducts a fair experiment on six real-world time series datasets, showing superior performance in out-of-distribution generalization.

**Weaknesses:**

I do not see any major weaknesses.

**Questions:**

Can this method be adapted for OOD time series regression and forecasting?

**Limitations:**

The authors adequately addressed the limitations.

---

> ### Author Rebuttal · Authors · 2024-08-07
>
> Thanks for recognizing our work.
>
> **(Q1)** Can this method be adapted for OOD time series regression and forecasting?
>
> **(Reply to Q1)** Yes, our proposed tri-level learning framework is designed to learn robust representations for time series OOD generalization, which means the learned representations could potentially be applied to other downstream tasks.

---

> > ### Comment · Reviewer_9773 · 2024-08-11
> >
> > Thanks for your response, after reading your response and other reviews, I keep the score unchanged.

---

### Official Review · Reviewer_tk3X · 2024-07-13

**Soundness:** 3
**Presentation:** 3
**Contribution:** 3
**Rating:** 5
**Confidence:** 2

**Summary:**

The paper explores the challenge of OOD generalization in time series data using pre-trained LLMs. The authors propose a novel framework TTSO that integrates both sample-level and group-level uncertainties. They also introduce a stratified localization algorithm tailored to this tri-level optimization problem, theoretically demonstrating its convergence. The paper presents extensive experiments to validate the method’s effectiveness.

**Strengths:**

1. The tri-level learning framework uniquely combines sample-level and group-level uncertainties, providing a comprehensive approach to OOD generalization.
2. The paper includes a solid theoretical foundation, with analyses that justify the proposed method and its iteration complexity.
3. The stratified localization algorithm offers a novel solution to the tri-level optimization problem, enhancing scalability and computational efficiency.
4. Extensive experiments on real-world datasets demonstrate the effectiveness of the proposed method, showing significant performance improvements.
5. The study leverages the advanced capabilities of LLMs in time series analysis, contributing to the emerging field of using foundational models for non-linguistic data.

**Weaknesses:**

1. The tri-level learning framework might be overly complex for practical applications, potentially limiting its adoption.
2. The proposed method, especially the stratified localization algorithm, may incur high computational costs, which could be a barrier for large-scale applications.
3. The paper could benefit from a more comprehensive comparison with other state-of-the-art methods in time series OOD generalization.
4. There is a need for more discussion on the real-world applicability and potential limitations of the proposed method in various domains.
5. Some sections of the paper are dense and challenging to follow, particularly the theoretical analyses, which might be difficult for a broader audience to understand.

**Questions:**

1. How does the tri-level learning framework handle scenarios with highly imbalanced time series data?
2. Can the proposed stratified localization algorithm be adapted for other types of data beyond time series?
3. What are the specific computational requirements for implementing the proposed method on large-scale datasets?

---

> ### Author Rebuttal · Authors · 2024-08-07
>
> **(W1 & W2)** The tri-level learning framework might be overly complex for practical applications, potentially limiting its adoption. The proposed method, especially the stratified localization algorithm, may incur high computational costs, which could be a barrier for large-scale applications.
>
> **(Reply to W1 & W2)** We appreciate the concern regarding the complexity of the tri-level learning framework. Time series OOD generalization is a challenging problem that necessitates a sophisticated and comprehensive approach. The TTSO framework is theoretically motivated by Theorem 2 (derived from [1]) and represents the **first work** to integrate both sample-level and group-level uncertainties under a tri-level learning framework for time series OOD generalization. This approach is crucial for effectively tackling the unique challenges of time series OOD generalization.
>
> Additionally, the stratified localization algorithm is **more computationally** efficient compared to hypergradient-based tri-level optimization methods[2,3]. Furthermore, the decomposable nature of cutting planes provides a promising pathway for distributed implementations of TTSO (e.g., ADBO[4], AFBO[5]), allowing the algorithm to be effectively applied to large-scale applications.
>
> **(W3)** The paper could benefit from a more comprehensive comparison with other state-of-the-art methods in time series OOD generalization.
>
> **(Reply to W3)** We have included comparisons with two time series OOD generalization methods: DFDG [6] and CCDG [7]. The results is detailed in following tables.
>
> HHAR
>
> | Target| A    | B  | C  | D   | AVG  |
> | --------- | ---- | ---- | ---- | ---- | ---- |
> | DIVERSIFY | 73.7 | 64.2 | 78.9 | 71.2 | 71.8 |
> | DFDG| 71.2 | 65.8 | 74.1 | 70.4 | 70.3 |
> | CCDG| 73.0 | 63.2 | 77.3 | 72.4 | 71.5 |
> | TTSO*| 77.6 | 67.3 | 80.6 | 69.9 | 73.9 |
>
> PAMAP
>
> | Target| A    | B | C| D    | AVG  |
> | --------- | ---- | ---- | ---- | ---- | ---- |
> | DIVERSIFY | 74.0 | 84.0 | 56.5 | 72.9 | 72.0 |
> | DFDG| 73.1 | 80.5 | 59.2 | 70.2 | 70.8 |
> | CCDG| 72.3 | 84.8 | 56.6 | 72.1 | 71.5 |
> | TTSO* | 78.5 | 89.6 | 61.4 | 75.0 | 76.1 |
>
> WEASAD
>
> | Target    | A    | B| C | D  | AVG  |
> | --------- | ---- | ---- | ---- | ---- | ---- |
> | DIVERSIFY | 57.6 | 73.0 | 72.6 | 57.1 | 64.6 |
> | DFDG | 49.8 | 71.6 | 71.1 | 50.7 | 60.8 |
> | CCDG | 54.5 | 70.5 | 69.8 | 54.1 | 62.2 |
> | TTSO* | 59.5 | 71.9 | 77.3 | 65.0 | 68.4 |
>
> **(W4)** There is a need for more discussion on the real-world applicability and potential limitations of the proposed method in various domains.
>
> **(Reply to W4)** Per your suggestion, we have added more discussion of TTSO framework on the real-world applicability and potential limitations in our manuscript (Appendix G & F) as follows.
>
> 1. Real-world Applicability. Our study on time series OOD generalization has significant potential in sensor-based applications, such as human activity and emotion recognition. For instance, our method can improve model robustness and accuracy against distribution shifts in healthcare, sports training, and abnormal behavior detection.
> 2. Potential Limitations. While our method demonstrates significant potential in time series OOD generalization, the ideas we propose are versatile and can be extended to more general settings. However, this may pose a challenge since different types of data have distinct sample-level uncertainties, which require reformulating the third-level optimization problem to effectively manage these uncertainties.
>
> **(W5)** Some sections of the paper are dense and challenging to follow, particularly the theoretical analyses, which might be difficult for a broader audience to understand.
>
> **(Reply to W5)** We understand that the theoretical sections of our paper may be challenging for some readers. However, these analyses are essential as they provide the necessary foundation and justification for our method's robustness and effectiveness. Per your suggestion, we've added explanations to make these sections more accessible to a broader audience in Appendix A.
>
> **(Q1)** How does the tri-level learning framework handle scenarios with highly imbalanced time series data?
>
> **(Reply to Q1)** In this paper, our primary focus is on time series OOD generalization, not on handling highly imbalanced data. However, it's worth mentioning that the proposed framework is flexible and effective. By incorporating established techniques such as data resampling (e.g., oversampling minority classes or undersampling majority classes) and synthetic data generation (e.g., SMOTE), TTSO framework can be adapted to address data imbalance issues effectively.
>
> **(Q2)** Can the proposed stratified localization algorithm be adapted for other types of data beyond time series?
>
> **(Reply to Q2)** Yes, the TTSO framework we proposed is versatile and applicable to various data modalities. However, our focus in this work was specifically on time series data. Focusing on time series data allows us to provide a comprehensive analysis that might not be achievable if we were to cover multiple data types simultaneously.
>
> **(Q3)** What are the specific computational requirements for implementing the proposed method on large-scale datasets?
>
> **(Reply to Q3)** Currently, the proposed method has been tested on a setup with two NVIDIA RTX 4090 GPUs and an Intel i9-14th generation processor. For large-scale datasets, the computational requirements will scale accordingly.
>
> [1] A theory of learning from different domains (ML 2010)
>
> [2] A gradient method for multilevel optimization (NeurIPS 2021)
>
> [3] Betty: An Automatic Differentiation Library for Multilevel Optimization (ICLR 2023)
>
> [4] Distributed distributionally robust optimization with non-convex objectives (NeurIPS 2022)
>
> [5] Provably Convergent Federated Trilevel Learning (AAAI 2024)
>
> [6] Robust domain-free domain generalization with class-aware alignment (ICASSP 2021)
>
> [7] Conditional Contrastive Domain Generalization for Fault Diagnosis (IEEE TIM 2022)

---

> ### Author Response · Authors · 2024-08-13
>
> Dear Reviewer tk3X,
>
> With the discussion period ending soon, I wanted to thank you for your valuable feedback on our paper. We have made revisions based on your suggestions, which have significantly improved our work.
>
> If you find that our revisions have addressed your concerns, we would greatly appreciate any additional feedback you may have.
>
> Thank you for your time and consideration.
>
> Best regards,
>
> Authors #16764

---

> > ### Comment · Reviewer_tk3X · 2024-08-14
> >
> > Thank you for the reply. I have read through your rebuttal. I will keep my score unchanged.

---

### Official Review · Reviewer_BvSM · 2024-07-13

**Soundness:** 3
**Presentation:** 2
**Contribution:** 2
**Rating:** 4
**Confidence:** 4

**Summary:**

The paper studies the problem of OOD generalization in time series tasks, building on recent observations that use data level uncertainties and group level uncertainties. This has been a successful way to build robust, transferrable representation. The paper additionally also includes an additional maximization for data augmentation that makes it the "tri-level" learning framework. The paper also theoretically analyzes generalization properties of the proposed algorithm.

The TTSO method is also used to fine-tune LLMs which is used for time series classification in OOD scenarios. In this part, it builds on some recent works that utilize pre-trained or fine-tuned LLMs in other domains, leveraging the superior feature learning capacities of these large frontier/foundation models.

**Strengths:**

* The paper addresses an important problem to build robust techniques for time series data, which is under studied and often quite challenging due to the inherent noise in time varying datasets.
* Using an LLM within a framework for generalization in time series appears to be a novel framing
* The paper does extensive theoretical and empirical analysis to demonstrate the properties and performance of TTSO.

**Weaknesses:**

On the surface of it, the paper shows a good improvement over several commonly used methods in OOD generalization, improving on different benchmarks. However, many aspects of the paper are unclear:
* Two of the main aspects of the paper are not sufficiently well motivated in my opinion --
	* (a) _OOD generalization in time series_: Why is this particular method effective for time series? as far as the assumptions made in the paper go, this is a general technique that can be applied to any arbitrary dataset/modality. While it is true that time series problems do not receive as much attention, that alone is not motivation enough. What assumptions in this paper restrict the use of TTSO for a broader set of modalities? If none, then how does this compare on other benchmarks?
	* (b) _LLMs for augmentation_: Why is the LLM necessary? The idea that LLMs can be used to produce different views of the sample is interesting, but the empirical performance appears to be too be marginal at best (+1.4% gain vs not using it from table 1). It doesn't seem to justify the additional computational burden for the small gain in performance. It would be helpful if the paper clearly articulates what the hypothesis is exactly here with regard to LLMs -- the ablation study in Fig 2 is a good start -- what kinds of LLMs benefit when fine-tuning? Is there a bias within the GPT-2 model used that is beneficial for these datasets? What about more advanced architectures/or LLMs trained on larger datasets, are they expected to give a bigger boost in performance or does it plateau?
* I think the paper is interesting and has a lot to say, but i would recommend making the core hypothesis and idea cleaner as I have outlined above.

**Questions:**

Please see above

**Limitations:**

Yes, it appears so.

---

> ### Author Rebuttal · Authors · 2024-08-07
>
> **(W1)** *OOD generalization in time series*: Why is this particular method effective for time series? as far as the assumptions made in the paper go, this is a general technique that can be applied to any arbitrary dataset/modality. While it is true that time series problems do not receive as much attention, that alone is not motivation enough. What assumptions in this paper restrict the use of TTSO for a broader set of modalities? If none, then how does this compare on other benchmarks?
>
> **(Reply to W1)** We appreciate your insightful comments. As pointed out by you, the OOD generalization for time series remains relatively under-explored. Our paper aims to bridge this gap by focusing on time series data, which allows us to provide a comprehensive analysis that might not be achievable if we were to cover multiple data types simultaneously.  However, please note its ability to be applied to other types of data as well is a testament to its versatility and robustness, rather than a limitation.
>
> In addition, given the extensive body of existing work on OOD generalization in computer vision and natural language processing, addressing such an extension within the scope of a single paper is impractical. We hope this work can spur further research in this area and lead to a more comprehensive understanding of OOD generalization via this tri-level learning across various data modalities.
>
> **(W2)** *LLMs for augmentation*: Why is the LLM necessary? The idea that LLMs can be used to produce different views of the sample is interesting, but the empirical performance appears to be too be marginal at best (+1.4% gain vs not using it from table 1). It doesn't seem to justify the additional computational burden for the small gain in performance. It would be helpful if the paper clearly articulates what the hypothesis is exactly here with regard to LLMs -- the ablation study in Fig 2 is a good start -- what kinds of LLMs benefit when fine-tuning? Is there a bias within the GPT-2 model used that is beneficial for these datasets? What about more advanced architectures/or LLMs trained on larger datasets, are they expected to give a bigger boost in performance or does it plateau?
>
> **(Reply to W2)** Thank you for your valuable feedback. Your feedback is invaluable in enhancing the quality of our work. The inclusion of an LLM within the proposed TTSO framework is **optional** rather than **mandatory**. Specifically, the TTSO framework can be effectively applied to time series OOD generalization without the integration of an LLM. The consideration of LLM within the framework is due to its potential as an emerging research direction that warrants further exploration. In fact, previous studies have demonstrated that LLMs have been shown to be effective in transfer learning across various modalities[1], and pre-trained transformers can improve OOD robustness [2,3]. Additionally, in cases where computational complexity presents a significant challenge, the TTSO framework may be employed without the use of an LLM to tackle time series OOD generalization.
>
> To further address your concerns regarding the hypothesis and efficacy of using LLMs in our framework, we conducted additional ablation experiments that focused on different LLM architectures and parameter sizes (e.g., base model and large model), including encoder-only (e.g., BERT), decoder-only (e.g., GPT-2), and encoder-decoder models (e.g., BART), to explore which configurations offer the most benefit during fine-tuning. The results are presented in the following table.
>
> | Architecture           | Version | HHAR | PAMAP | WESAD | AVG  |
> | ---------------------- | ------- | ---- | ----- | ----- | ---- |
> | Encoder-Only (BERT)    | Base    | 64.3 | 66.9  | 64.4  | 64.2 |
> |                        | Large   | 61.7 | 52.5  | 62.3  | 58.8 |
> | Decoder-Only (GPT)     | Base    | 72.9 | 76.1  | 68.4  | 72.5 |
> |                        | Large   | 64.5 | 69.4  | 66.5  | 66.8 |
> | Encoder-Decoder (BART) | Base    | 57.3 | 65.4  | 64.2  | 62.3 |
> |                        | Large   | 55.5 | 61.2  | 61.4  | 59.4 |
>
> The results indicate that decoder-only architectures, specifically the GPT-2 base model in this experiments, achieve the best performance. However, increasing the number of parameters in all three architectures leads to a significant drop in performance across 3 architectures for time series OOD generalization.
>
> To further explore how the number of parameters affects performance, we conducted experiments using GPT-2 models with varying numbers of Transformer layers on 3 datasets to evaluate their OOD generalization performance. For these experiments, we utilized 20% of each dataset. As shown in Figure 3 (in the attached PDF), the results demonstrate that optimal OOD generalization performance is achieved with a configuration of 8 Transformer layers. Based on this findings, we incorporate this optimal layer configuration with TTSO framework, yielding improved results as detailed in the following table.
>
> | Target | A    | B    | C    | D    | AVG   |
> | ------ | ---- | ---- | ---- | ---- | ----- |
> | HHAR   | +1.4 | +0.1 | +0.8 | +0.9 | +0.80 |
> | PAMAP  | +1.0 | -1.4 | +0.8 | -0.3 | +0.03 |
> | WEASAD | +5.5 | -3.5 | +2.5 | -0.1 | +1.35 |
>
> [1] One fits all: Power general time series analysis by pretrained lm (NeurIPS 2023)
>
> [2] How Good Are Large Language Models at Out-of-Distribution Detection? (arXiv 2023)
>
> [3] Pretrained Transformers Improve Out-of-Distribution Robustness (ACL 2021)

---

> > ### Comment · Reviewer_BvSM · 2024-08-12
> > **Acknowledgement of Rebuttal**
> >
> > Thank you for your clarifications and additional analysis.
> > (a) Regarding time series, I agree versatility is a strength, but the reason for studying time-series OOD generalization needs to be motivated better other than it is under studied. There is no clear reasoning for why this method is suitable for time series, which have their own sets of challenges and problems from noisy data, rate variation, length variation etc. different from images/text problems.
> > (b) The fact that LLMs are optional cannot be an argument when it is in the title of the paper -- it is indeed a strength that LLMs can be incorporated, but this is not motivated sufficiently enough. The performance does not show a clear benefit, and the new results seem to further make this a bit muddled since we see larger models doing poorer than smaller ones, which is pretty much counter to current intuition on scaling LLMs.
> >
> > Given these issues, I will maintain my score as it is.

---

> ### Author Response · Authors · 2024-08-13
>
> >  (a) Regarding time series, I agree versatility is a strength, but the reason for studying time-series OOD generalization needs to be motivated better other than it is under studied. There is no clear reasoning for why this method is suitable for time series, which have their own sets of challenges and problems from noisy data, rate variation, length variation etc. different from images/text problems.
>
> **Reply to (a)** Thank you for your insightful comments. We agree that time series data presents unique challenges, such as noise, rate variation, and length variation. This type of data is crucial in many real-world applications, including healthcare, sports training, and abnormal behavior detection. Despite its significance, OOD generalization in time series has been relatively under-explored. To fill this gap, we have introduced the TTSO framework, which is **theoretically grounded** and specifically designed to tackle both sample-level and group-level uncertainties inherent in time series data. The novelty of our approach lies in how it models these uncertainties under a **tri-level learning** framework, providing a robust solution for time series OOD generalization.
>
> Thank you for agreeing that time series OOD generalization problem is under-explored and this constitutes a motivation of our study. Please note that imposing strong assumptions on time series data is impractical and will inevitably compromise the effectiveness of the proposed method since time series data comes in diverse formats and properties, such as periodic versus non-periodic patterns, regular versus irregular sampling, and stationary versus non-stationary behavior etc. Developing a general OOD generalization framework for time series data is already a significant challenge in itself. Finally, as previously stated, our primary research area is time series learning. Concentrating on the time series OOD generalization allows us to delve deeply into this specific problem and conduct thorough analysis and experiments.
>
> While temporal order is an inherent characteristic of all time series data, the presence of other properties—such as seasonality, trends, and irregular sampling—is not universal. These characteristics vary depending on the specific context and nature of the time series. For instance, some time series may exhibit pronounced seasonal patterns or trends, while others may be stationary with no significant autocorrelation. The diverse formats and characteristics of time series data motivate us to develop a versatile and flexible framework (TTSO) that can effectively address these variations. In summary, TTSO is particularly well-suited for time series data due to its ability to accommodate the diverse and variable properties inherent in such data.
>
> > (b) The fact that LLMs are optional cannot be an argument when it is in the title of the paper -- it is indeed a strength that LLMs can be incorporated, but this is not motivated sufficiently enough. The performance does not show a clear benefit, and the new results seem to further make this a bit muddled since we see larger models doing poorer than smaller ones, which is pretty much counter to current intuition on scaling LLMs.
>
> **Reply to (b)** Thank you for your constructive suggestion. Larger models are generally expected to perform better due to their enhanced capacity to capture complex patterns, which aligns with the established scaling laws observed in natural language models. OpenAI's research on scaling laws[1] provides substantial evidence that, within the domain of **natural language**, larger models tend to exhibit better performance when scaled **appropriately** in terms of model size, dataset size, and computational resources.
>
> However, please note that these scaling laws are not **universally applicable** across all domains. In the context of time series data, the applicability of scaling laws remains an **open question**. The characteristics of time series data differ significantly from those in natural language, and there is **no theoretical guarantee** that scaling laws will always hold in this domain. This uncertainty highlights a key area for future research, as understanding whether and how scaling laws apply to time series data could yield valuable insights.
>
> As we all agree that the TTSO framework is versatile and while it has been thoroughly studied in this paper for time series data, the proposed framework has the potential to be applied to other types of data beyond time series. In summary, our work opens up new avenues for OOD generalization through the introduction of a tri-level optimization framework. We hope that this innovative approach will inspire further research and development in the area of OOD generalization.
>
> Moreover, if the inclusion of LLMs in the title of the paper is a concern, we are open to considering its removal, if allowed. We appreciate your feedback on this matter.

---

### Official Review · Reviewer_Z4zT · 2024-07-14

**Soundness:** 4
**Presentation:** 3
**Contribution:** 4
**Rating:** 7
**Confidence:** 4

**Summary:**

Out-of-Distribution（OOD）generalization in ML emphasizes on improving model adaptability and robustness aginst unseen and potentially adversarial data. This paper explores OOD generalization for time series data with pre-trained Large Language Models and proposes a novel tri-level learning framework to handle the data distribution uncertainties. It goal is to address not only sample-level but also group-level uncertainties in the new dataset. The paper offers a theoretical analysis to justify the method and develops a cutting plane strategy for the tri-level optimization problem. It demonstrates guaranteed convergence. Extensive experiments on real-world datasets confirm the effectiveness and efficiency of the proposed method.

**Strengths:**

1) OOD generation is an important problem in ensuring the robustness and reliability of machine learning models. It becomes increasingly important in AI as machine learning models and systems are expected to be deployed in numerous real-world applications in the near future.
2)The tri-level learning framework is grounded in robust theoretical principles and offers a fresh perspective for modeling and studying the OOD problem. This framework opens new avenues for investigating OOD challenges in time series data.
3)The TTSO algorithm is proven to converge, with the authors also determining its convergence speed.
4)Extensive experimental studies have been carried out to validate the effectiveness of the proposed methods.

**Weaknesses:**

While the paper is well organized overall, the clarity of this paper can be further enhanced via providing examples to illustrate the concepts of sample-level and group-level uncertainties in time series.

**Questions:**

Can the TSSO framework be employed to manage data uncertainties in other type of data, for example natural language data?

---

> ### Author Rebuttal · Authors · 2024-08-07
>
> **(W1)** While the paper is well organized overall, the clarity of this paper can be further enhanced via providing examples to illustrate the concepts of sample-level and group-level uncertainties in time series.
>
> **(Reply to W1)** Thank you for your insightful comments. Per your suggestion, we have provided an example using the HHAR dataset to illustrate the concepts of sample-level and group-level uncertainties in the attached PDF (global rebuttal). Specifically, we use the x-axis values from accelerometer data collected by the ‘samsungold_1’ device from four users.
>
> In Figure 1, sample-level uncertainty is shown by plotting time series data from a specific label (e.g., 'walking'), where each line represents a different time window. The variations among these lines illustrate the inherent noise, which represents sample-level uncertainty.
>
> Figure 2 demonstrates the group-level uncertainty by displaying the distribution of x-axis values from the accelerometer across different groups (users). Each color represents a distinct group, and each group's unique characteristics contribute to the overall group-level uncertainty.
>
> **(Q1)**  Can the TTSO framework be employed to manage data uncertainties in other type of data, for example natural language data?
>
> **(Reply to Q1)** Yes, the TTSO framework we proposed is versatile and applicable to various data modalities. However, our focus in this work was specifically on time series data. Focusing on time series data allows us to provide a comprehensive analysis that might not be achievable if we were to cover multiple data types simultaneously.

---

### Author Rebuttal · Authors · 2024-08-07

Figure 1, Figure 2 and Figure 3 is in the attached PDF.

---

### Author Response · Authors · 2024-08-13
**Summary of Contributions and Modifications**

We sincerely thank the Area Chairs for managing the review process and the reviewers for their thoughtful consideration and evaluation of our work. To facilitate further discussions, we have summarized our contributions and modifications as follows:

**Contributions**

* We propose a novel **tri-level learning** framework, TTSO, to tackle time series OOD generalization with pre-trained LLMs. By integrating both sample-level and group-level uncertainties, TTSO advances beyond traditional single or bi-level methods in OOD generalization. We provide a theoretical analysis based on the Vapnik-Chervonenkis dimension to elucidate **generalization properties** of TTSO, and demonstrate its effectiveness through extensive real-world datasets.
* We develop a stratified localization method using cutting planes, which is **more computationally** efficient compared to hypergradient-based tri-level optimization methods. And the decomposable nature of cutting planes provides a promising pathway for distributed implementations of TTSO, allowing the algorithm to be effectively applied to large-scale applications.
* We conducted a **thorough theoretical analysis** of the proposed algorithm. The proposed stratified localization algorithm for tri-level optimization shows guaranteed convergence and we theoretically derive that the iteration complexity of the proposed algorithm for achieving an $\epsilon$-stationary point is bounded by $\mathcal{O}(\frac{1}{\epsilon^2})$.

**Modifications**

* (Clarity Enhancements) We have included examples using the HHAR dataset to better illustrate the concepts of sample-level and group-level uncertainties in time series data (Appendix H).
* (Discussion on Applicability and Limitations) We have added more discussion of TTSO framework on the real-world applicability and potential limitations (Appendix G & F).
* (Ablation Experiments) To clarify the efficacy of LLMs in our framework, we conducted additional ablation experiments, the results of which are now included in the revised manuscript (Appendix I).
* (More Baselines) We have expanded our experimental analysis to include further comparisons with more time series OOD generalization methods in the revised manuscript (Section 5).
* (Explanations to Theorem) We have added more explanations to make theoretical sections more accessible to a broader audience (Appendix A).

We hope these revisions will address the concerns raised and better demonstrate the contributions of our work.

---

### Decision · Program_Chairs · 2024-09-25

**Decision:**

Accept (poster)

**Comment:**

This paper focuses on time-series out-of-distribution generalization with LLMs, provides new methods that use uncertainties and localization, and gives theoretical analysis on the generalization properties and convergence. Reviewers appreciated the application of LLMs to the problem domain and found good improvements in the experimental section. Although the theoretical work may not be easily accessible to a wide audience, most of the reviewers appreciated the work to show convergence properties.

While the paper restricted its focus to time series and did not test on other modalities, reviewers noted that the methods did not seem to rely on any specific aspects of time-series problems, and in that sense could have been tested more broadly.

In the discussion phase, the authors showed new results using a variety of architectures and LLM sizes. Somewhat curiously, larger LLMs led to worse performance on the paper’s tasks. The authors are encouraged to think more about the possible causes of this observation and provide substantive discussion, as a practitioner could pick up this technique, think to use the most advanced LLM available, and find poor OOD generalization results.